# Identification of New Sub-Fossil Diatoms Flora in the Sediments of Suncheonman Bay, Korea

Mirye Park [1], Sang Deuk Lee [2,*], Hoil Lee [3], Jin-Young Lee [4], Daeryul Kwon [1] and Jeong-Min Choi [5]

1   Protist Research Team, Microbial Research Department, Nakdonggang National Institute of Biological Resources (NNIBR), 137, Donam 2-gil, Sangju-si 37182, Korea; mirye@nnibr.re.kr (M.P.); kdyrevive@nnibr.re.kr (D.K.)
2   Bioresources Collection and Research Team, Bioresources Collection and Bioinformation Department, Nakdonggang National Institute of Biological Resources (NNIBR), 137, Donam 2-gil, Sangju-si 37182, Korea
3   Center for Active Tectonics, Geology Division, Korea Institute of Geoscience and Mineral Resources (KIGAM), 124, Gwahak-ro, Yuseong-gu, Daejeon 34132, Korea; hoillee@kigam.re.kr
4   Geological Research Center, Geology Division, Korea Institute of Geoscience and Mineral Resources (KIGAM), 124, Gwahak-ro, Yuseong-gu, Daejeon 34132, Korea; jylee@kigam.re.kr
5   Department of Suncheon Bay Preservation, Suncheon-si 58027, Korea; haema@korea.kr
*   Correspondence: diatom83@nnibr.re.kr; Tel.: +82-54-530-0898; Fax: +82-54-0899

**Abstract:** Suncheonman Bay, Korea's most representative estuary, is an invasive coastal wetland composed of 22.6 km$^2$ of tidal flats surrounded by the Yeosu and Goheung Peninsulas. In January 2006, this region was registered in the Ramsar Convention list in Korea, representing the first registered wetland. Estuaries are generally known to have high species diversity. In particular, several studies have been conducted on planktonic and epipelic diatoms as primary producers. Suncheonman Bay has already been involved in many biological and geochemical studies, but fossil diatoms have not been evaluated. Therefore, we investigated fossil diatoms in Suncheonman Bay and introduced sub-fossil diatoms recorded in Korea. One sedimentary core has been extracted in 2018. We identified 87 diatom taxa from 52 genera in the SCW03 core sample. Of these, six species represent new records in Korea: *Cymatonitzschia marina*, *Fallacia hodgeana*, *Navicula mannii*, *Metascolioneis tumida*, *Surirella recedens*, and *Thalassionema synedriforme*. These six newly recorded diatom species were examined by light microscopy and scanning electron microscopy. The ecological habitats for all the investigated taxa are presented.

**Keywords:** sub-fossil diatom; sediment; Suncheonman Bay; new record

## 1. Introduction

An estuary can be defined as a semi-enclosed coastal body of water that has a free connection with the open ocean, within which seawater is diluted with freshwater derived from land drainage [1,2]. River mouths, coastal bays, tidal marsh systems, and sounds all fit this definition. Estuaries are transitional zones between freshwater and marine habitats. Due to tides and storms, the water level and salinity vary in estuaries [3]. They are most commonly located in low-relief coastal regions. Estuaries and wetlands are among the most productive aquatic ecosystems, providing a home for both freshwater and marine plants, and a source of nutrients for a variety of animal communities adapted to brackish waters [4,5]. Moreover, they filter out pollutants supplied to the ocean [4,6,7]. Thus, many animals rely on estuaries that have abundant species diversity for food, places to breed, and migration stopovers [8,9] (https://oceanservice.noaa.gov/facts/estuary.html (accessed on 8 February 2021)).

In South Korea, where three sides are surrounded by the sea, there are numerous estuaries such as the Nakdonggang, Keumgang, and Seomjingang. Coastal wetlands cover approximately 2800 km$^2$, which represents approximately 3% of the total land area [10–12]. Suncheonman Bay, Korea's most representative estuary, is an invasive coastal wetland

composed of 21.6 km$^2$ of tidal flats and 5.4 km$^2$ of reed fields surrounded by the Yeosu and Goheung Peninsulas [10]. In January 2006, it became the first registered wetland in the Ramsar Convention list in Korea, as it was designated as a "wetland protected area" by the Ministry of Land, Transport and Maritime Affairs in December 2003 [10,13]. In June 2008, Suncheonman Bay was designated as national cultural property, "Myeongseung" number 41, and in 2010, south-western tidal flats in Korea, including Suncheonman Bay (south-western coast tidal flats), were included in the UNESCO World Heritage Tentative List. Such registrations and designations demonstrate the recognition of its ecological and environmental value [14] (https://whc.unesco.org/en/tentativelists/5482 (accessed on 20 April 2021)). In particular, the natural environment and ecosystems within Suncheonman Bay are well preserved, making them a habitat favorable for many species of marine organisms [15]. In these regions, primary producers such as diatoms play a critical role as a food source for large invertebrates and fishes [16].

Diatoms are unicellular algae characterized by a biomineralized (opaline) cell wall that may fossilize and be preserved in the sedimentary record [17,18]. The sub-fossil diatoms herein described consist of Holocene diatom remains not fully involved in the fossilization process. Diatoms thrive in very different environments (e.g., hot springs, polar regions, and fresh, brackish, and marine waters) and are extremely sensitive to physical and chemical changes (e.g., temperature, salinity, and nutrients) in the water [18–23]. Therefore, fossil diatoms represent an excellent source of information about past climate change and its effect on aquatic ecosystems. There have been relatively few studies on sub-fossil diatoms along the southern Korean coast. Marine to brackish sub-fossil diatom assemblages were initially studied in the Pohang and Gampo sediments in 1975 [24], then they were extended to the regions of Bukpyeong [25] and Pohang [26,27] in the East Sea and the regions of the Mankyung-Dongin river estuary [28], Dodaecheon River [29], Ilsan estuary [30], Chollipo [31], Isanpo [32], and Sabsi-do and Kunsan in the Yellow Sea [33].

The Suncheonman Bay has been used to study various environmental characteristics, including grain size and organic matter in tidal flat sediments [10], seasonal water quality, pollution, environmental safety [13,34], and other geochemical characteristics, as well as local inhabitants such as benthic invertebrates, plants, fishes, birds, bacteria, and fungi [6,14,35–39]. Among the studies conducted to date, the investigation of phytoplankton communities in the Dong Cheon River and Isa Cheon River stream into Suncheonman bay was the most interesting [40]. In this study, we describe a newly recorded sub-fossil diatom assemblage recovered in the sediments of the Suncheonman Bay.

## 2. Materials and Methods

### 2.1. Coring and Sampling of Sediment

Drilling was carried out using a peat core sampler (52 mm diameter; Peat Sampler, Eijkelkamp Soil & Water, Giesbeek, Netherlands). One sediment core (= SCW03 with a length of 6.0 m) was retrieved from Suncheonman Bay in Korea on 11 June 2018 (Figure 1, Table 1). The core was transported to the laboratory after vacuum packing in a plastic bag to prevent drying and oxidation. Sedimentological description and subsampling were performed after the sediment core profile was cut vertically in half [41]. Shell fragments in the SCW03 core were selected for the analysis of chronology and diatoms because the sediment layer was well preserved.

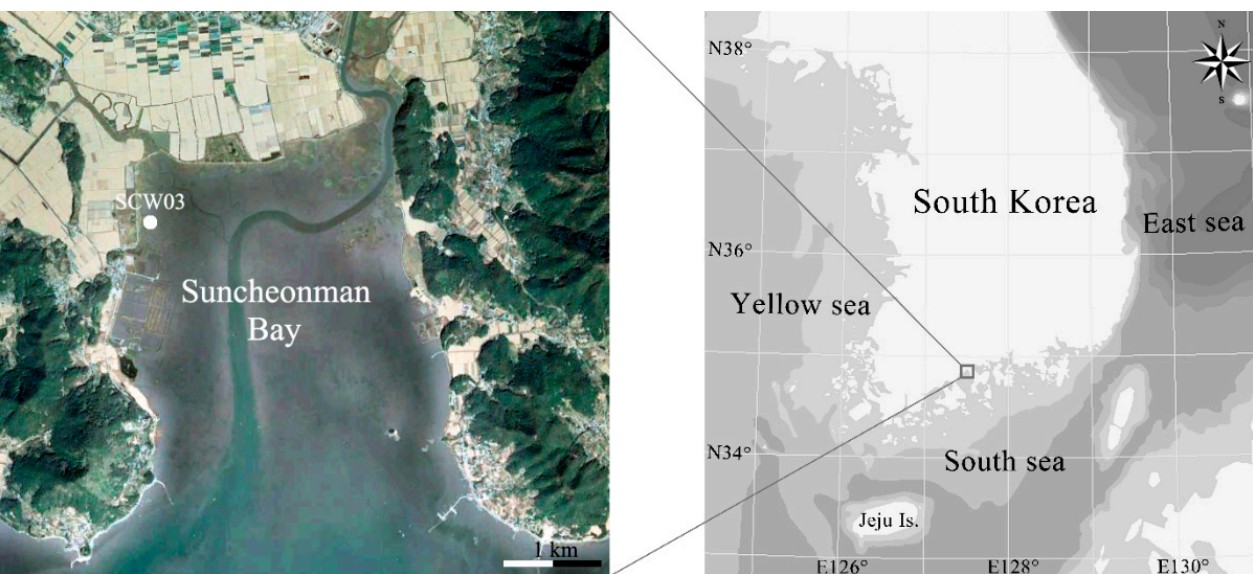

**Figure 1.** Sampling sites in Suncheonman Bay.

**Table 1.** Information of sampling sites.

| Site | Location | Depth (m) | Latitude (N) | Longitude (E) |
|---|---|---|---|---|
| SCW03 | Haksan-ri Byeollyang-myeon Suncheon-si, Jeollanam-do | 6 | 34°52′10.1266″ | 127°29′27.5667″ |

### 2.2. Analysis of Chronology

Age dating was performed using an accelerator mass spectrometer (AMS) at the Korean Institute of Geoscience and Mineral Resources (KIGAM), Korea. The estimated ages were calibrated by the OxCal statistical analysis program (http://c14.arch.ox.ac.uk (accessed on 20 May 2021)).

### 2.3. Sample Preparation for Diatom Identification

Thirteen samples of diatoms were collected every 0.5 m along the SCW03 core. Their analysis was conducted according to the following steps: 1 g of sediment was dried at 60 °C for 24 h; the siliceous material was boiled with 20 mL of 30% hydrogen peroxide ($H_2O_2$) and washed with distilled water to remove organic matter; the treated samples were mounted with Pleurax (Mountmedia, Wako, Japan) and briefly heated using an alcohol lamp for subsequent analysis using a light microscope (LM; Eclipse Ni, Nikon, Tokyo, Japan). Photomicrographs were taken using a digital camera (DS-Ri2, Nikon, Tokyo, Japan). Some remaining peroxide-cleaned samples were filtered using 2.0-μm polycarbonate membrane filters (Nuclepore, Whatman, Maidstone, UK). The membranes were placed on stubs and coated with gold-palladium for analysis using a field emission scanning electron microscope (FE-SEM; MIRA 3, TESCAN, Brno-Kohoutovice, Czech Republic). SEM photomicrographs of all the samples were used to identify the diatoms. Morphological analyses of diatoms were performed using ImageJ v1.32 software (NIS-Elements BR4.50.00, Nikon, Tokyo, Japan) [42]. Taxonomical nomenclature was based on recent taxonomic information guidelines [43].

## 3. Results

### 3.1. Sedimentary Facies Analysis

The core SCW03 mostly consists of greenish-grey silty clay and can be divided distinguished into three sedimentary facies according to color, fossils content, and sedimentary structure (Figure 2A,C). Facies A is characterized by yellowish-brown mottling structures. Shell fragments are not observed in this facies. In Facies B, the yellowish-brown mottling

structures are less abundant and shell fragments are observed to occur sporadically. The size of shell fragments are about 2 mm in diameter. Facies C is represented by highly concentrated shells and shell fragments in several-centimeters-thick intervals.

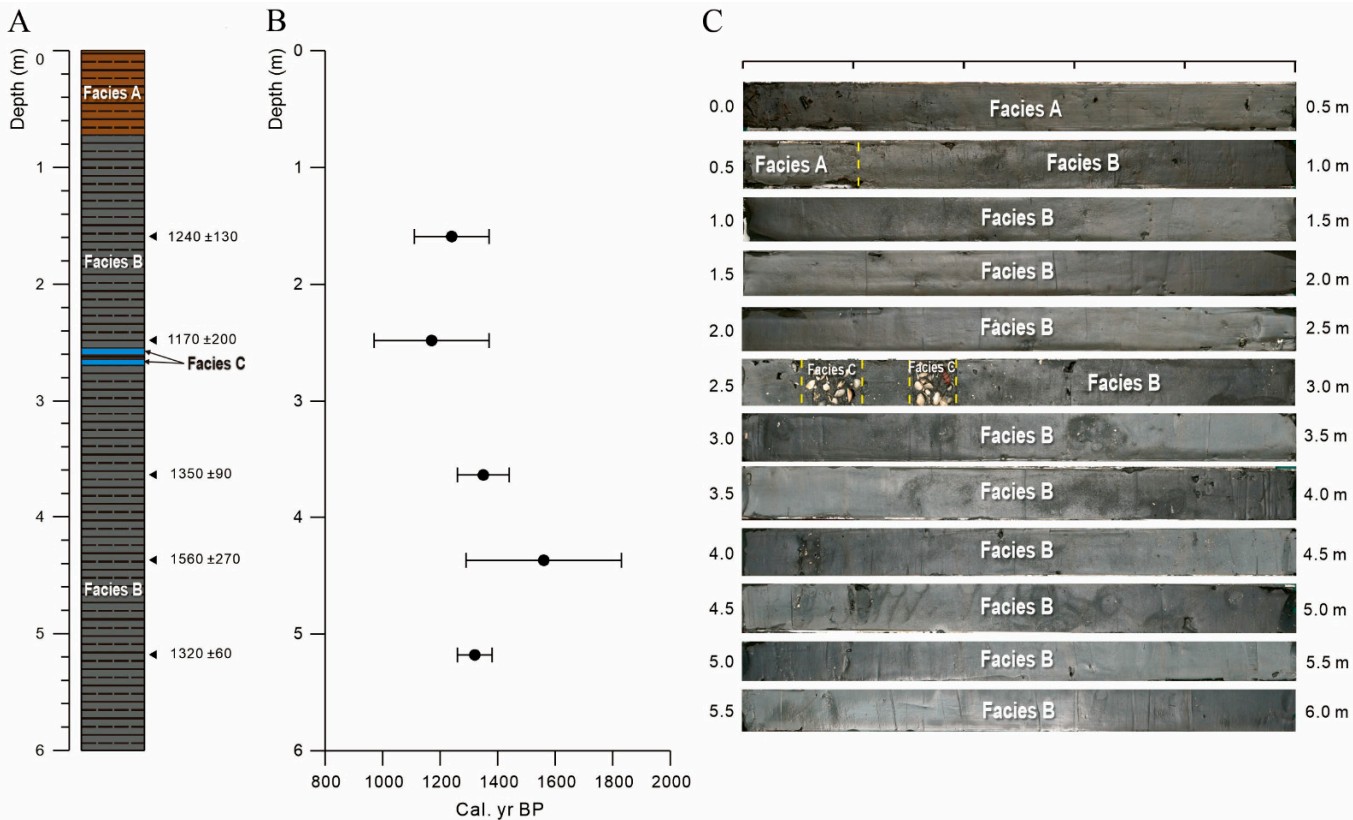

**Figure 2.** (**A**) Stratigraphic section, (**B**) result of age dating, and (**C**) photographs of core SCW03.

The sediments of core SCW03 are interpreted as deposited in a tidal flat [44,45]. It is very similar to the current tidal flat environment. Abundant mottling structures in facies A originate from an oxygen-rich environment, indicating that the sea level has gradually decreased slightly in facies B. In the meantime, highly concentrated shells and shell fragments of facies C are interpreted as sedimentation caused by flooding or storm events [45].

### 3.2. Age Dating

The results of age dating for five samples in the core SCW03 shows a range from 1170 to 1560 cal. yr BP (Table 2 and Figure 2B).

**Table 2.** Results of AMS $^{14}$C dating and calibrated dates for core SCW03.

| Depth (m) | Elevation (m) | $^{14}$C yr BP ($\pm 1\sigma$) | Cal. yr BP ($\pm 2\sigma$) | Laboratory Code | Dating Material |
|---|---|---|---|---|---|
| 1.59 | 0.07 | $1340 \pm 20$ | $1240 \pm 130$ | ITg180525 | Shell fragments |
| 2.48 | $-0.82$ | $1240 \pm 20$ | $1170 \pm 200$ | ITg180526 | Shell fragments |
| 3.64 | $-1.98$ | $1480 \pm 20$ | $1350 \pm 90$ | ITg180527 | Shell fragments |
| 4.37 | $-2.71$ | $1660 \pm 20$ | $1560 \pm 270$ | ITg180528 | Shell fragments |
| 5.18 | $-3.52$ | $1410 \pm 20$ | $1320 \pm 60$ | ITg180529 | Shell fragments |

### 3.3. Diatom Assemblages

A total of 87 diatom species belonging to 52 different genera were identified in the sediments from Suncheonman Bay in Korea (Table 3); of these, six species, namely, *Cymatonitzschia marina, Fallacia hodgeana, Navicula mannii, Metascolioneis tumida, Surirella recedens,* and *Thalassionema synedriforme,* were never described before in this area. The diatom flora

encountered in this survey was composed of 87 taxa, which were classified into 52 genera. We present information on the valve shape, occurrence depth in sediment, and habitat of 87 species, including the six newly recorded species. Information collected about the newly recorded diatoms identified here included taxonomic information, illustrations, basionyms, synonyms, original description references, depth in the core, distribution, and diagnosis (Table 3).

**Table 3.** Occurrence (black squares) and habitat of diatom species by depth. A total of 72 diatom species belonging to 52 different genera were identified. Star marks on the specific name are newly recorded species in Korea (6 species: *Cymatonitzschia marina*, *Fallacia hodgeana*, *Navicula mannii*, *Metascolioneis tumida*, *Surirella recedens*, and *Thalassionema synedriforme*).

| | SCW03 (m) | 0.1 | 0.5 | 1.0 | 1.5 | 2.0 | 2.5 | 3.0 | 3.5 | 4.0 | 4.5 | 5.0 | 5.5 | 6.0 | Habitat | Reference |
|---|---|---|---|---|---|---|---|---|---|---|---|---|---|---|---|---|
| 1 | *Achnanthes* sp. | | | | | | ■ | | | | | | | | | - |
| 2 | *Actinocyclus octonarius* | | | ■ | | | | | | ■ | | | | | marine | [46] |
| 3 | *Actinoptychus senarius* | | | ■ | ■ | ■ | | | | | ■ | | ■ | ■ | marine | [46] |
| 4 | *Amphora pediculus* | | | ■ | ■ | ■ | | ■ | ■ | ■ | | | | | freshwater | [47] |
| 5 | *Amphora* sp. | ■ | | | | | | | | | | | | | | - |
| 6 | *Auliscus sculptus* | ■ | | | | | | | | | | | | | marine | [48] |
| 7 | *Bacillaria paxillifera* | | | | | | | | | ■ | | | | | marine | [49] |
| 8 | *Bacteriastrum* sp. | | | | | | | | | ■ | ■ | | | | | - |
| 9 | *Chaetoceros affinis* | ■ | ■ | ■ | ■ | ■ | | ■ | | | ■ | ■ | | ■ | marine | [50] |
| 10 | *Chaetoceros compressus* | | | | | | | | ■ | | | | | | marine | [51] |
| 11 | *Chaetoceros lorenzianus* | | | | | | | | | | | ■ | | | marine | [52] |
| 12 | *Chaetoceros* sp. | | ■ | | | ■ | | | ■ | ■ | ■ | ■ | ■ | ■ | | - |
| 13 | *Chaetoceros resting spores* | | | ■ | ■ | ■ | | ■ | ■ | ■ | | ■ | ■ | ■ | | - |
| 14 | *Climaconeis mabikii* | | | | ■ | | | | | | | | | | marine | [53] |
| 15 | *Cocconeis placentula* | ■ | | | | | ■ | | | ■ | | ■ | | | freshwater | [54] |
| 16 | *Cocconeis* sp. | | | | | | | | | ■ | | | ■ | | | - |
| 17 | *Coscinodiscus asteromphalus* | | | | | | | | | ■ | | ■ | | | marine | [55] |
| 18 | *Coscinodiscus centralis* | | | | | | | | ■ | | ■ | | | | marine | [56] |
| 19 | *Coscinodiscus radiatus* | ■ | ■ | ■ | ■ | | ■ | ■ | ■ | ■ | ■ | | ■ | ■ | marine | [57] |
| 20 | *Coscinodiscus* sp. | | | | | | | | | | | | ■ | | | - |
| 21 | *Cyclotella litoralis* | ■ | ■ | ■ | | ■ | ■ | ■ | ■ | ■ | ■ | ■ | ■ | ■ | marine/freshwater | [58] |
| 22 | *Cyclotella meneghiniana* | | | | | | | | | | | | ■ | | marine/freshwater | [59] |
| 23 | *Cyclotella ocellata* | | | | | | | ■ | ■ | ■ | | | | | freshwater | [60] |
| 24 | *Cyclotella* sp. | | | ■ | | | | | | | | | | | | - |
| 25 | *Cymatonitzschia marina** | | | | ■ | | | | | | | | | | marine | [61–66] |
| 26 | *Cymatosira lorenziana* | | | | | | | | | | | | ■ | | marine | [67] |
| 27 | *Cymatotheca* sp. | | | | | | | | ■ | | | | | | | - |
| 28 | *Cymatotheca weissflogii* | | | ■ | | | | | | ■ | ■ | | | | marine | [68] |
| 29 | *Delphineis* sp. | ■ | | | | | | | | | | | | | marine | - |
| 30 | *Diploneis elliptica* | ■ | ■ | ■ | ■ | ■ | ■ | ■ | | ■ | | | | | marine/freshwater | [69] |
| 31 | *Diploneis* sp. | ■ | ■ | ■ | | | | | | ■ | ■ | | | | | - |
| 32 | *Diploneis weissflogii* | | | ■ | | ■ | | ■ | | ■ | ■ | | | | marine | [70] |
| 33 | *Discostella stelligera* | | | | | | | | | | ■ | | | | freshwater | [71] |
| 34 | *Ditylum sol* | | | ■ | | | | | | | | | | | marine | [72] |
| 35 | *Encyonema* sp. | | | ■ | | | | | | | | | | | | - |
| 36 | *Epithemia adnata* | ■ | | | | | | | ■ | | | | | | freshwater | [73] |

**Table 3.** *Cont.*

| | SCW03 (m) | 0.1 | 0.5 | 1.0 | 1.5 | 2.0 | 2.5 | 3.0 | 3.5 | 4.0 | 4.5 | 5.0 | 5.5 | 6.0 | Habitat | Reference |
|---|---|---|---|---|---|---|---|---|---|---|---|---|---|---|---|---|
| 37 | *Fallacia hodgeana** | | | | | | | | | | | | | ■ | Freshwater/brackish water | [74] |
| 38 | *Fallacia* sp. | | | | | ■ | | | | | | ■ | | ■ | | - |
| 39 | *Fragilaria capucina* | ■ | | | | | | | | | | | | | marine/freshwater | [75] |
| 40 | *Frustulia vulgaris* | | | | | | | ■ | | ■ | | | | | freshwater | [76] |
| 41 | *Giffenia cocconeiformis* | | | | | | ■ | | | ■ | | | | | marine | [70] |
| 42 | *Giffenia* sp. | ■ | ■ | ■ | ■ | ■ | ■ | ■ | ■ | ■ | ■ | ■ | ■ | ■ | | - |
| 43 | *Gomphonema* sp. | | | | ■ | | ■ | | | | | | | | | - |
| 44 | *Gyrosigma accuminatum* | ■ | ■ | | | | | ■ | | | | | | | freshwater | [77] |
| 45 | *Gyrosigma fasciola* | | | | | ■ | | | | | | ■ | | | marine | [78] |
| 46 | *Gyrosigma* sp. | | | | ■ | ■ | ■ | | | | ■ | ■ | | | | - |
| 47 | *Gyrosigma turgidum* | | | | | | | | | ■ | | | ■ | | marine | [79] |
| 48 | *Halamphora latecostata* | ■ | | | | | | | | | | | | ■ | freshwater | [80] |
| 49 | *Haslea ostrearia* | | | ■ | | | ■ | | | ■ | | | | | marine | [81] |
| 50 | *Hyalodiscus subtilis* | | | | ■ | | | | | | | | | | marine | [82] |
| 51 | *Lyrella* sp. | | | | | | | | | ■ | | | | | | - |
| 52 | *Navicula* sp. | ■ | ■ | | | ■ | ■ | ■ | ■ | | | ■ | | ■ | | - |
| 53 | *Navicula viridulacalcis* | | | | | ■ | | ■ | | | | | | | freshwater | [83] |
| 54 | *Navicula mannii** | | | | | | ■ | | | | | | | | brackish water/marine | [70] |
| 55 | *Nitzschia sigma* | ■ | | ■ | | ■ | | ■ | | | | ■ | | | brackish | [84] |
| 56 | *Nitzschia* sp. | ■ | ■ | ■ | | | | ■ | | ■ | ■ | ■ | | ■ | | - |
| 57 | *Paralia sulcata* | ■ | ■ | ■ | | ■ | ■ | ■ | ■ | ■ | ■ | ■ | ■ | ■ | marine | [85] |
| 58 | *Parlibellus delognei* | | | | | ■ | ■ | ■ | | ■ | ■ | | | | marine | [86] |
| 59 | *Petrodictyon gemma* | | | | | ■ | | | | | | | | | marine | [87] |
| 60 | *Petroneis marina* | ■ | ■ | ■ | | ■ | | | | ■ | | ■ | | | marine | [88] |
| 61 | *Pinnularia* sp. | | | | | ■ | | | | | | | | | | - |
| 62 | *Pleurosigma aestuarii* | | | | ■ | ■ | | ■ | ■ | ■ | | | | ■ | marine | [89] |
| 63 | *Pleurosigma diverse-striatum* | | | | | | | | | | | ■ | | | marine | [90] |
| 64 | *Pleurosigma normanii* | | | | | ■ | | ■ | | | | | | ■ | marine | [91] |
| 65 | *Pleurosigma* sp. | | | | | | | | | | | | ■ | | | - |
| 66 | *Pseudonitzschia pungens* | ■ | | | | | | | | | ■ | | ■ | | marine | [92] |
| 67 | *Rhaphoneis* sp. | | | | | | | | | | | ■ | | | | - |
| 68 | *Rhizosolenia setigera* | | | | ■ | | | | | | | ■ | | | marine | [81] |
| 69 | *Metascolioneis tumida** | | | ■ | ■ | ■ | | | | | | | | | marine | [93] |
| 70 | *Sellaphora americana* | | | | | | | | | | | ■ | | | freshwater | [94] |
| 71 | *Semiorbis* sp. | | | | | | | | | | | | ■ | | | - |
| 72 | *Surirella recedens** | | | | | | ■ | | | | | | ■ | | marine | [95,96] |
| 73 | *Surirella* sp. | | | ■ | ■ | ■ | | ■ | ■ | ■ | | ■ | | ■ | | - |
| 74 | *Thalassionema nitzschioides* | | ■ | ■ | | ■ | | ■ | | ■ | | ■ | | ■ | marine | [97] |
| 75 | *Thalassionema synedriforme** | | | | | | | ■ | | | | | | | marine | [98] |
| 76 | *Thalassiosira decipiens* | | | ■ | | | ■ | | | ■ | | | | | marine | [99] |
| 77 | *Thalassiosira eccentrica* | | | | ■ | | | ■ | | | ■ | ■ | ■ | ■ | marine | [99] |

**Table 3.** *Cont.*

| | SCW03 (m) | 0.1 | 0.5 | 1.0 | 1.5 | 2.0 | 2.5 | 3.0 | 3.5 | 4.0 | 4.5 | 5.0 | 5.5 | 6.0 | Habitat | Reference |
|---|---|---|---|---|---|---|---|---|---|---|---|---|---|---|---|---|
| 78 | *Thalassiosira oestrupii* | | | ■ | | ■ | | | | ■ | | | | ■ | marine | [100] |
| 79 | *Thalassiosira* sp. | | ■ | ■ | | ■ | ■ | ■ | | ■ | ■ | | ■ | ■ | | - |
| 80 | *Trachyneis aspera* | | | | | | | | | | | | ■ | | marine | [101] |
| 81 | *Triceratium dubium* | | | | | | | ■ | | | | | | | marine | [102] |
| 82 | *Tryblionella acuminata* | ■ | | | ■ | | | | ■ | ■ | | | | | marine | [103] |
| 83 | *Tryblionella coarctata* | | | | | ■ | | | | ■ | | | ■ | | marine | [104] |
| 84 | *Tryblionella granulata* | | | | | | | | | ■ | | ■ | | | marine | [70] |
| 85 | *Tryblionella punctata* | | | | | | | ■ | | ■ | | ■ | | ■ | marine/freshwater | [105] |
| 86 | *Tryblionella* sp. | | | ■ | | ■ | | | | ■ | | ■ | | | | - |
| 87 | *Tryblioptychus cocconeiformis* | | | | | | | ■ | | | ■ | ■ | ■ | ■ | marine | [106] |
| | Total | 21 | 14 | 23 | 19 | 31 | 17 | 27 | 14 | 34 | 20 | 27 | 19 | 30 | | |

*Cymatonitzschia marina* (F.W.Lewis) Simonsen 1974 (Figure 3A,B)
Basionym: *Cymatopleura marina* F.W.Lewis 1861 [107]
Synonym: *Cymatopleura marina* F.W.Lewis 1861 [107]
Original description: Simonsen 1974: 56, pl. 41: Figures 5–9 [108]
Description: Valves are observed to be solitary, usually lying in the valve view. Valves are linear lanceolate, with very acute ends. Valves are strictly isopolar, and not constricted in the middle. Overall dimensions include a valve length ranges from 58.42 to 67.94 μm and valve width from 9.14 to 11.28 μm. The valve faces have numerous undulations (9–11), with a distance between two undulations in the ranges from 4.81 to 7.97 μm. Undulations are found to have a nearly trapezoidal shape (Figure 3A). The valve surface has irregular punctate on the undulations. A raphe system is observed running around one side of the valve margin. Striae uniseriate are found to be densely spaced, with approximately 28–29 per 10 μm observed, on one side of the valve margin (Figure 3B).
Depth occurrence in the core: 2.0 m.
Distribution: This species is reported from brackish water to marine environments mainly [61–66]. This taxon is reported from some estuary, e.g., East River, New York and Long Island Sound [109]. *Cymatopleura marina* was first recorded from the Indian Ocean [108].
Differential diagnosis: This genus differs from *Cymatopleura*. The genus *Cymatopleura*, as a member of the Surirellaceae, has a completely different raphe morphology, which runs along the edge of the valve around the entire margin, whereas in *Cymatonitzschia* it is, as in *Nitzschia*, limited to one of the sides [108]
Remarks of raphe: *Cymatonitzschia marina* has an eccentric keeled raphe placed through the edge of the valve, and it appears on one of the sides [108,110].
*Fallacia hodgeana* (R.M.Patrick and Freese) Y.H.Li and H.Suzuki 2014 (Figure 3C–F)
Basionym: *Navicula hodgeana* R.M.Patrick and Freese 1961 [111]
Synonym: *Navicula hodgeana* R.M.Patrick and Freese 1961 [111]
Original description: Li et al., 2014 in p. 33 [74]
Description: Valves are observed to be solitary, usually lying in the valve view. Valves are naviculoid and linear-elliptic, with bluntly rounded ends. Overall dimensions include length ranges from 14.73 to 15.42 μm and width ranges from 4.21 to 4.61 μm. The valve face is nearly flat with a slightly curved raphe (Figure 3C,D, arrow). Central raphe endings are proximately hooked (Figure 3E,F, arrowheads). Terminal raphe fissures exhibit a sickle-shaped curve in the same direction. Some parts of the striae are covered with a thin siliceous covering, or conopeum, on the external valve surface. Tow slits opening of the canal, present near the terminal raphe fissures, are also observed (Figure 3F, arrowhead). Areolae are found to be curved upward and were directly connected to the mantle. The

finely porous conopeum extends outward from the outer edge of the raphe sterna, running through the surface, and connect to the proximal edge of the mantle. Numerous peg-shaped structures are observed in the nonporous margin of the conopeum along the proximal edge of the mantle. The elongated areolae, with an approximate length and width of 0.28–0.42 µm and 0.15–0.18 µm, respectively, are found on the hyaline area of the valve surface with an undulated junction. The elongated areolae were found in groupings of 12 per 5 µm. Peg-shaped structures, in groups of 12 per 5 µm, are also observed, finely porous on the conopeum 12–13 per 1 µm transversely.

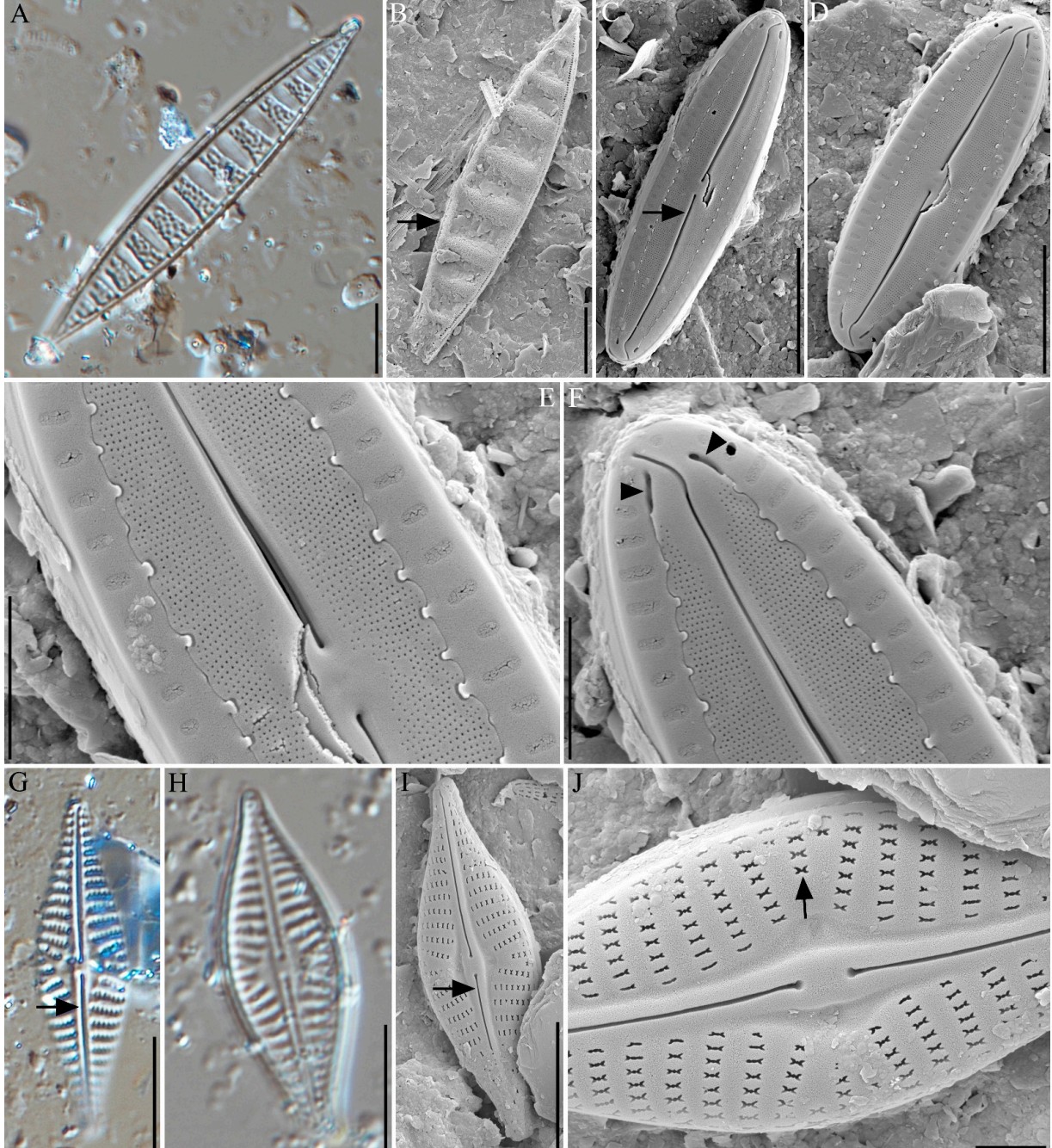

**Figure 3.** Light microscope (**A**,**G**,**H**) and field emission scanning electron microscopic (**B**–**F**,**I**,**J**) photomicrographs of diatoms: (**A**,**B**) *Cymatonitzschia marina* (the arrow points to raphe), (**C**–**F**) *Fallacia hodgeana* (in **C** the arrow points to raphe; in **F** the arrow points to the terminal openings), (**G**–**J**) *Navicula mannii* with raphe and ribbon-shaped areola (the arrow points to raphe and ribbon-shaped areola). Scale bar = 10 µm.

Depth occurrence in the core: 6.0 m.

Distribution: This species lives in fresh- to brackish-water environments. It was first reported from scrapings of small rock submerged at the edge of a lagoon as *Navicula hodgeana* [111]. Li et al. 2014 collected this species from Edogawa River, Japan. This taxon is known to benthic diatom [74,112,113].

Differential diagnosis: *Fallacia hodgeana* possesses morphological features such as a single H-shaped plastid, depressed lateral sterna interrupting striae that contain round areolae enclosed by hymen; well-developed, finely porous conopeum; and a canal system between the primary silica layer and the conopeum. These features indicate that this species does not belong to the genus *Navicula* or *Pseudofallacia* [74]. This species is related to *Navicula dissipata*. The length-to-breadth ratio is similar, although *N. dissipata* is a larger taxon. The clear central area is narrower in our taxon, and the striae are composed of many fine puncta instead of a few large ones. The median ends of the raphe are close together as in *N. dissipata* [111].

Remarks of raphe: *Fallacia hodgeana* has a slightly curved raphe that terminates at fissures curved in the same way. The distal end fissures were sickle-like and curved in the same direction (Figure 3C arrow). The central endings lie close to each other, and slightly curved slits seem to be promoted from the general valve [74].

*Navicula mannii* Hagelstein 1939 (Figure 3G–J)

Synonym: *Navicula elegantissima* Meister 1935 [114]

Original description: Hagelstein 1939, p. 388, pl. 7, Figures 7 and 8 [115]

Description: Valves are observed to be solitary. Valves are broadly lanceolate, and abruptly constricted toward the ends (Figure 3G–I). Overall dimensions include average length and width ranges from 28.28 to 30.09 μm and from 8.57 to 9.44 μm, respectively. The axial area is narrow, and becomes gradually wider, larger, and rounded toward the central area (Figure 3I). Raphe is observed to be very slightly curved filiform style with a very thickened and hyaline sternum (Figure 3I,J, arrow). Striae are very coarse and of low density (9–11 in 10 μm); they are observed to strongly radiate in the middle and then become parallel towards the ends. The central area striae alternate between longer and shorter forms (longer striae 4 areolae, and shorter striae 2 areolae). Areolae are observed to be ribbon-shaped, and approximately 5–6 are found in 2 μm sections (Figure 3J, arrow).

Depth occurrence in the core: 2.7 cm.

Distribution: *Navicula mannii* was reported in brackish water or marine environments [70,116,117]. Navarro (1983) reported the taxon in tropical temperate waters from the southwestern coast of Puerto Rico [116]. This species was known to neritic, pantropical, and cosmopolitan [116]. Ohtsuka (2005) collected the species from a muddy tidal flat in the Ariake Sea in south-western Japan [117].

Differential diagnosis: Hagelstein (1939) described that the *Navicula mannii* have minutely punctate areolae, but we found the ribbon-shaped areolae on the striae based on SEM observation in this study [115]. Ultrastructural studies of this species are rarely performed using a SEM; this study represents the first example of this approach.

Remarks of raphe: *Navicula mannii* has a straight raphe. Proximal raphe ends have an expanded pore-like shape and bent distal raphe ends (Figure 3G,H,J, arrows). Normally, *Navicula* spp. have a straight raphe system, unlike the raphe shapes of *Navicula cryptocephala* and *Navicula gregaria*, which commonly occur in Korea. *N. mannii* and *N. cryptocephala* both have drop-like internal ends, but *N. mannii* has a more pore-like end than *N. cryptocephala* [83,118], with a T-shaped structure. In contrast, *Navicula gregaria* has a different raphe shape than *N. mannii*, which is bent in the same direction as the raphe and exhibits asymmetrical thickening, beside the proximal raphe ends and beside the raphe rib [118].

*Metascolioneis tumida* (Brébisson ex Kützing) Blanco and Wetzel 2016 (Figure 4A,B)

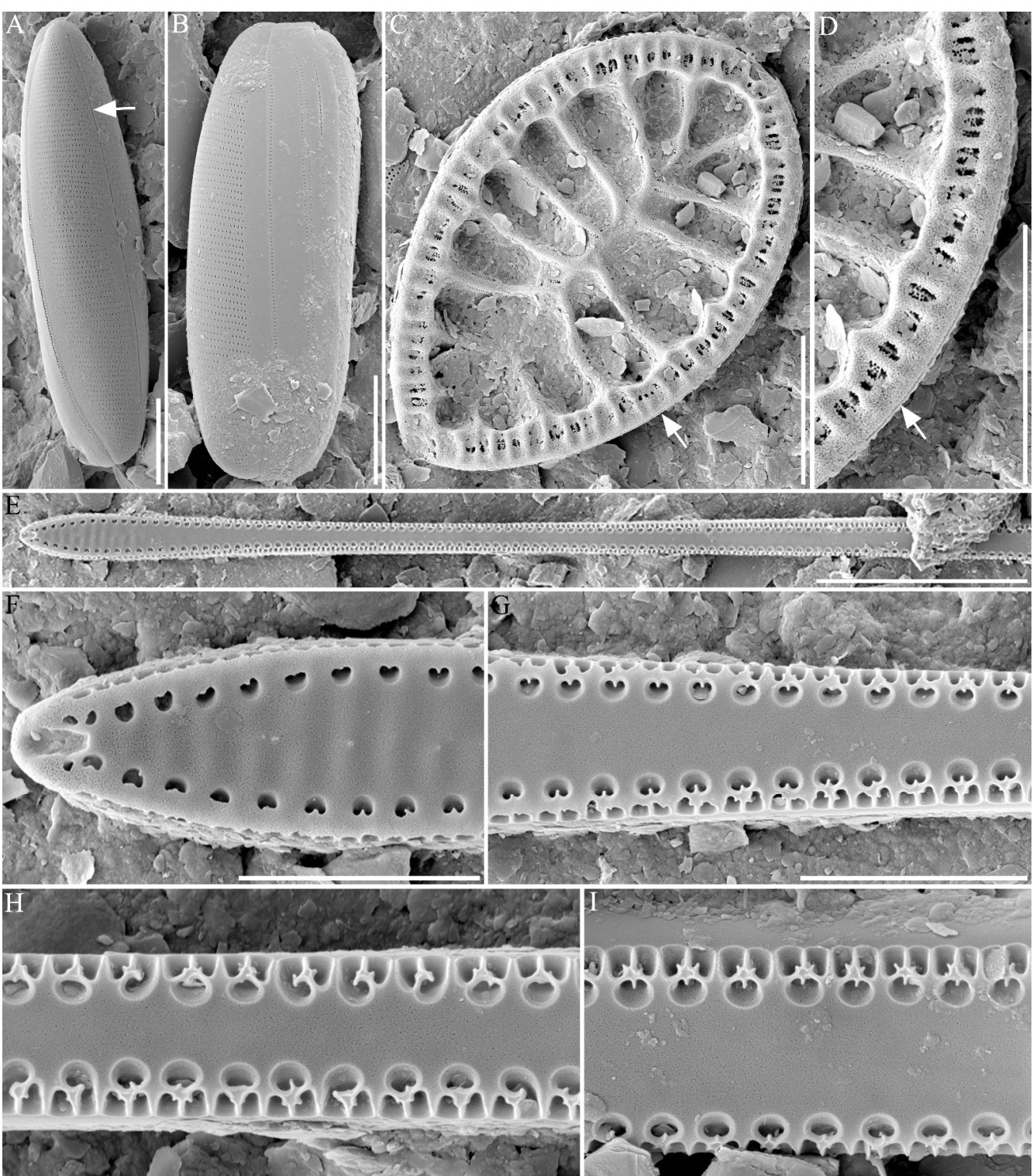

**Figure 4.** Field emission scanning electron microscopic photomicrographs of diatoms. (**A**,**B**) *Metascolioneis tumida* (the arrow points to raphe), (**C**,**D**) *Surirella recedens* (the arrow points to raphe), (**E**–**I**) *Thalassionema synedriforme*. Scale bar = 10 µm.

Basionym: *Navicula tumida* Brébisson ex Kützing 1849 [119]
Synonym: *Navicula tumida* Brébisson ex Kützing 1849 [119]
*Scoliopleura tumida* (Brébisson ex Kützing) Rabenhorst 1864 [120]
*Microstigma tumida* (Brébisson) Meister 1919 [121]
*Scoliotropis tumida* (Brébisson ex Kützing) R.M.Patrick and Freese 1961 [111]
*Scolioneis tumida* (Brébisson ex Kützing) D.G.Mann 1990 [122]

*Navicula jenneri* W.Smith 1853 [84]

Scoliopleura jenneri Grunow 1860 [123]

Original description: Blanco, S. and Wetzel, C.E., 2016, pp. 195–205 [124].

Description: Valves are found to be solitary with one-layered valves. Valves are linear lanceolate with bluntly rounded apices (Figure 4A). Overall dimensions include an average length and width ranges from 41.72 to 52.5 μm and from 6.05 to 6.50 μm, respectively. Cells are usually found in a girdle view and twisted about the apical axis (Figure 4B). The valve mantle is relatively deep, and its face curved moderately into mantles. Striae uniseriate (16–17 in 10 μm) with small poroids (12–15 μm in 5 μm) are observed. The raphe system is twisted and sigmoidal in shape (Figure 4A, arrow). The raphe sternum is generally narrow and slightly wider in thcenterre. The raphe is found to be straight with simple raphe endings and straight terminal fissures extending to the valve margin. The girdle consists of several open bands. The band closest to the valve bears one transverse row of poroids 31–32 at 10 μm.

Depth occurrence in the core: 1.0, 1.5, 2.0 m.

Distribution: This species has been found in marine habitats. Stoermer et al. (1999) presented this taxon in a checklist of diatoms as *Navicula tumida* from the marine environment in the Laurentian Great Lakes [125]. Vilicic et al. (2002) reported the species as *Scoliopleura tumida* from the eastern Adriatic Sea [93]. This species was listed to the British marine diatoms as *Scoliopleura tumida* [126,127]. Méléder et al. (2007) reported the taxon as *Scolioneis tumida* in a sediment of mudflat from Bourgneuf Bay, France [128].

Differential diagnosis: Formerly included in *Scolipleura* taxon, but lacking the offset central raphe endings and longitudinal canals of that genus. Distinguishable from *Scoliotropis* by having fewer plastids, which lie against the valves rather than the girdle, by the simple uniseriate striae and raphe structure [122].

Remarks of raphe: *Metascolioneis tumida* (syn. *Scolioneis tumida*, *Navicula tumida*) has a slightly twisted raphe and raphe sternum that is normally narrow and becomes expanded in the center (Figure 4A, arrow). External central raphe endings have straight fissures along the valve margin. Furthermore, internal central endings are T-shaped and elongated [122].

*Surirella recedens* A.W.F.Schmidt 1875 (Figure 4C,D)

Homotypic synonym: *Surirella fastuosa* var. *recedens* (A.W. F. Schmidt) Cleve 1878 [129]

*Surirella fastuosa* var. *typica* f. *recedens* (A. W. F. Schmidt) Deby 1897 [130]

Original description: Schmidt and Fricke 1875, pls 17–20. [131]

Description: Valves are found to be solitary, strongly silicified, and lying in the valve or girdle view. Valves are heteropolar with a broadly rounded headpole and cuneate footpole. Overall dimensions include a length and width of 35.08 μm and 22.83 μm, respectively. The valve surface has four costae of 10 μm in length. The valve margin includes fibulae, siliceous braces, 7–8 fibulae, 10 μm. The raphe system runs around the entire valve margin and is located within a canal (Figure 4C,D; arrows). The canal is raised above the valve's surface. One or more potulae are located between the two fibulae. Four or more fibulae are located between the two costae.

Occurring depth in Core: 2.7, 5.5 m.

Distribution: This taxon was known to marine species [95,96]. López-Fuerte and Siqueiros-Beltrones (2016) reported the species as a benthic diatom from coastal waters in Mexico [132] and the Nanaura mudflat in Ariake Sea, Japan [117]. However, *Surirella recedens* was found in brackish waters from Cochin Backwater south in the Indian Ocean [133].

Differential diagnosis: *S. recedens* is composed of a heteropolar valve, whereas *Surirella fastuosa* has an isopolar valve [134]. *S. recedens* is smaller overall, in comparison to *S. fastuosa*, and more lanceolate in shape. In addition, *S. fastuosa* has more noticeable apices than *S. recedens* [135]. Goldman (1990) identified the *Surirella* cf. *fastuosa* based on the outline, length, infundibula, and circular pattern of the valve [136].

Remarks of raphe: *Surirella recedens* (Syn. *Surirella fastuosa* var. *recedens*, *Surella fastuosa* var. *typical* f. *recedens*) *Surirella* sp. has a raphe that is located along the margin of the valve (Figure 4D, arrow). A raphe positioned within a canal might be elevated above the

valve surface in several species [137]. *S. recedens* shows a representative *Surirellaceae* raphe system, positioned along the margin of the valve (Figure 4D, arrow).

*Thalassionema synedriforme* (Greville) G.R.Hasle 1999 (Figure 4E–I)

Basionym: *Asterionella synedriformis* Greville 1865 [138]

Synonym: *Asterionella synedriformis* Greville 1865 [138]

*Thalassionema javanicum* Grunow Hasle in Hasle and Syvertsen 1996 [139]

Depth occurrence in the core: Hasle, G.R. 1999, pp. 54–59, 23 figures [140].

Description: The valves are heteropolar spatulate, linear, long, slightly wider in the middle, and constricted towards the head pole, rather than towards the foot pole (Figure 4E). Valve width ranges from 2.66 to 2.87 μm in the middle part of the valve and from 4.03 to 4.40 μm in the middle part of the footpole. An apical spine is located in the head pole, not the foot-pole part of the valve. The valve face is flat, with a wide sternum and slight undulation (5–6 in 5 μm) at the foot pole (Figure 4F). Areolae are placed within the valve face and valve mantle. Areolae are heart-shaped (5–6 in 5 μm) near the foot pole part, but similar to Y- or flower-shaped occlusions (5–6 in 5 μm) toward the middle part of the valve (Figure 4F–I). Labiate processes are placed at each pole of the valve. An opening of the labiate process places the external apex in the foot pole [141].

Depth occurrence in the core: 3.0 m.

Distribution: *Thalassionema synedriforme* was known to marine species. Hasle (2001) mentioned the species is restricted in tropical and subtropical waters [98]. This species was recorded for the first time from Argentinean coastal waters [142].

Differential diagnosis: *Asterionella synedriformis* Greville is the basionym of *Thalassionema synedriforme* [98]. Frenguelli (1941) mentioned that he found valves with 9–10 areolae within 10 μm and illustrated a linear, fragmentary specimen that, according to the valve outline and areola density, might also be attributed to *Thalassionema frauenfeldii*, but not to *T. synedriforme* [143]. Additionally, in the case of *Thalassiothrix javanica* (Grunow), Hustedt and Frenguelli illustrated a specimen that was slightly heteropolar with 6–7 areolae within 10 μm, which differs in areolae density and valve outline from *Thalassionema synedriforme* (12–16 areolae in 10 μm, according to Hasle 2001) [98].

## 4. Discussion

### 4.1. Diatoms in SCW03

In this study, the analysis of sub-fossil diatoms among core samples obtained from Suncheon Bay, Korea, was carried out. Within the SCW03 core sample, a total of 52 genera and 87 species of sub-fossil diatoms were identified, with locations ranging from the surface to within 6 m of the basement, and among them, six species of newly recorded sub-fossil diatom never recorded before were found. At a depth of 4.0 m, the maximum variety of diatom samples was observed, with 23 genera 34 species, while depths of 0.5 m and 3.5 m revealed the lowest variety of diatoms, with 12 genera 14 species and 10 genera 14 species, respectively (Table 3). The highest and lowest taxonomic richness occur at depths of 4.0 m (23 genera, 34 species) and 3.5 m (10 genera, 14 species), respectively (Table 3).

Among the diatoms observed, *Amphora* sp., *Auliscus sculptus,* and *Fragilaria capucina* were found only at 0.1 m depths; therefore, they are hypothesized to have only recently entered the Suncheonman Bay (Table 3). *Cymatosira lorenziana, Fallacia hodgeana, Pleurosigma* sp., *Semiorbis* sp., and *Trachyneis aspera* are no longer observed since they appeared to only occur at 6.0 m depth. It is hypothesized that this is due to climate, environmental, or topographical changes in the Suncheonman Bay habitat. Intriguingly, *C. lorenziana* is mainly found in warm waters, whereas *T. aspera* is mainly distributed in the Antarctic area, with almost opposite habitat characteristics; however, the causative environmental changes in Suncheonman Bay could not be identified in this study [144–146]. Nevertheless, identifying past environmental changes in Suncheon Bay is an important source of information for the prediction of future environmental changes, and should be investigated further.

Most of the marine and brackish species occur at a depth of 5.0 to 6.0 m of the SCW03 core sample, and the emergence of freshwater species gradually increased within depth

ranges of 3.0 to 4.5 m. Interestingly, only marine and brackish species appeared at the 2.5 m section (Table 3). Marine and some freshwater species appeared within occur the range of 1.0 m to 2.0 m depths, while only marine and brackish species appeared at 0.5 m depth; finally, freshwater species increased again at 0.1 m depth. These changes in the flora of diatoms assemblage composition observed suggest that the sediments recovered from SCW03 were deposited in a marine area that was scarcely influenced by freshwater inflow in the past (3.0–4.5 m depth; about 1260–1830 yr BP) and experienced a renewed influx of fresh water in recent times (0.1 m depth; about $1340 \pm 20$ yr BP) (Tables 2 and 3, Figure 2) [147,148]. More accurate results and solid interpretations of the triggers responsible for such environmental changes require further studies of comprehensive environmental change, including a quantitative analysis of diatoms and a chronological analysis of core samples.

### 4.2. New Recorded Taxa from Korea

In this study, we identified six species newly recorded in Suncheonman Bay area: Cymatonitzschia marina, Fallacia Hodgeana, Navicula mannii, Metascolioneis tumida, Surirella recedens, and Thalassionema synedriforme. N. mannii was first identified by light microscopy in 2005; however, in this study, we observed the ultrastructure of N. mannii and discovered a ribbon-shaped areolae by Fe-SEM [117]. The ribbon-shaped areolae of N. mannii are the first to ever be recorded.

Unrecorded sub-fossil diatoms in Suncheonman Bay were discovered at 1.0 m depths. This study is therefore meaningful because if we study the modern composition of diatoms in Suncheonman Bay or Korea, we would estimate six previously unrecorded diatoms were extinct in this area.

**Author Contributions:** Data curation, formal analysis, writing—original draft, and writing—review and editing, M.P., D.K., and S.D.L.; funding acquisition, S.D.L.; field investigation, J.-Y.L. and J.-M.C.; writing—review and editing, M.P., S.D.L., H.L., J.-Y.L., D.K., and J.-M.C. All authors have read and agreed to the published version of the manuscript.

**Funding:** This work was supported by grants from the Nakdonggang National Institute of Biological Resources (NNIBR) (NNIBR202101108) projects.

**Institutional Review Board Statement:** Not applicable.

**Informed Consent Statement:** Not applicable.

**Data Availability Statement:** Not applicable.

**Acknowledgments:** Thanks to Seung Won Nam, Suk Min Yun, and Pyo Yun Cho for helping with the coring of sediment.

**Conflicts of Interest:** The authors declare no conflict of interest.

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
