# Peer review of "Identification of New Sub-Fossil Diatoms Flora in the Sediments of Suncheonman Bay, Korea"

_jmse, doi:10.3390/jmse9060591_

Round 1
Reviewer 1 Report
The novelty is insufficient to support a qualified research paper in JMSE. Currently, the only novel finding is a couple of diatoms that are firstly recorded in Korea. However, how this finding impacts or shapes our understanding on Korean coastal ecosystem is missing. We discover new thinks every day, but the finding itself is neutral and not every new thing deserves a publication in academia. Therefore, It is the authors' duty to illustrate clearly the impact of the findings in terms of ecology, genetics, physiology or any topics interest the authors.
Another choice for the fate of this manuscript is to publish this finding in journals of platforms on taxonomy that are specific to recording new discovered species (not a new species) in a region.
Author Response
Point 1: The novelty is insufficient to support a qualified research paper in JMSE. Currently, the only novel finding is a couple of diatoms that are firstly recorded in Korea. However, how this finding impacts or shapes our understanding on Korean coastal ecosystem is missing. We discover new thinks every day, but the finding itself is neutral and not every new thing deserves a publication in academia. Therefore, It is the authors' duty to illustrate clearly the impact of the findings in terms of ecology, genetics, physiology or any topics interest the authors.
Another choice for the fate of this manuscript is to publish this finding in journals of platforms on taxonomy that are specific to recording new discovered species (not a new species) in a region. 

Response 1: We submitted special issue, ‘Marine biology’, ‘Taxonomy and Ecology of Marine Algae’ section. This reason, we thought this manuscript is suitable for the journal scope. According to the comments of the reviewers and editor, the manuscript was rewritten by adding the results of AMS dating analysis. Please we would like you to review the rewritten manuscript.

Reviewer 2 Report
The manuscript submitted by Park and collegues and titled 'Identification of New Sub-fossil Diatoms and Flora on the Sediments of Suncheonman Bay, Korea', provides interesting results about the Korean diatom record, that can be relevant for further studies concerning past climate reconstructions. Therefore, the manuscript surely deserve to be published in the Journal of Marine Science and Engineering. However, I strongly suggest some major revisions, listed below.
I am convinced that the authors will be able to address the points highlighted. Then, I will be glad to read once again this interesting paper.
TITLE
Why do you cite the term 'Flora'? In the manuscript you only describe the diatom content of the sediment core, not other floral elements. Maybe 'Diatom flora', but not 'Diatoms and Flora'.
Be careful to use the term 'on' ('on the Sediments'): if I correctly understood, you found the diatoms 'in' the sediments.
ABSTRACT
Lines 21-23: What is the difference between 'fossil' and 'sub-fossil' diatoms? As far as I remember a sub-fossil is a biological remain that was not involved in all the fossilization steps (e.g., no diagenesis), but I am not completely sure. I suggest to define it more clearly and to use just one of the two terms throughout the manuscript. But the major problem is: how old is your diatom assemblage? Without any age constraint, how can you apply your nice findings for past climate reconstructions? You should try to address this point in the manuscript. Even if you cannot provide absolute ages you can give some estimates…
KEYWORDS
Line 29: 'new record' is probably better than 'new recorded'.
INTRODUCTION
Line 38: what is the meaning of 'geologically transient'? Moreover, I do not find a clear reason for the use of the term 'therefore' immediately after the previous sentence, since there is no logical connections between geologically transient environments and their productivity.
Lines 41-42: do you mean that estuaries provide a 'sink' for the excess of organic matter produced and consumed at the land-to-sea transition zone? I suggest to rephrase the sentence…
Lines 59-60: add 'them' after 'making'; add 'favorable' after 'habitat'; delete 'ranging'; and specify what are the macrobenthic and the macrorganisms you cite. Why note to specify the most important species of macrobenthic and macroorganisms involved? It is just a minor suggestion…
Lines 60-62: I suggest to better emphasize the role of diatoms. On my opinion is better to state 'In these regions, primary producers like diatoms play a critical role as a food source for large invertebrates and fishes'.
Lines 63-67: I suggest to rephrase the first sentence in this way: 'Diatoms are unicellular algae characterized by a biomineralized (opaline) cell wall that may fossilize and be preserved in the sedimentary record.' Then you should emphasize that 'Diatoms thrives in very different environments (e.g., hot springs, polar regions, fresh, brackish and marine waters) and are extremely sensitive to the physical and chemical changes (e.g. temperature, salinity and nutrients) of the waters." Then, on my personal opinion you must highlight the logical link between the two sentences, for example stating 'Therefore, fossil diatoms represent an excellent source of information about past climate change and its effect on the aquatic ecosystems'.
Lines 69-72: not necessary, you already state at lines 60-62 that diatoms are critical in these environments.
Lines 72-73: not necessary.
Lines 72-78: not necessary and wrong in many parts. Note that the Cretaceous ranges from 145 to 65 Ma, that diatom probably originated in the Triassic, and most importantly, that the fossil diatom record is not at all restricted to the Miocene (that lasted from around 23 to 5 Ma). I suggest to delete this passage.
Line 83: what is a 'sedimentary' diatom? Fossil (or subfossil if you prefer) is ok, but I never heard the term 'sedimentary diatoms'.
Lines 90-91: please rephrase, not clear… 'has been used to freshwater flow'???
Lines 92-94: please rephrase. I suggest: 'In this study we describe a newly recorded sub-fossil diatom assemblage recovered in the sediments of the Suncheonman Bay'.
MATERIALS AND METHODS
Line 99: 'lateral' instead of 'horizontal'; 'sediments' instead of 'sedimentary layers'.
Lines 101-102: how these cores correlates among each others? I suggest to provide some pictures of these cores, highlighting their most relevant sedimentological features (e.g., color, sedimentary structures etc.).
Line 105: see above. Where is this 'sedimentological description'?
Lines 106-107: what do you mean for 'analysis of chronology'?; what do you mean for 'sediment layer was well preserved'?
Table 1: 'Location' instead of 'Address'; I think 'Depth' is better than 'Altitude'.
Lines 111-112: 'were collected every 0.5 m along the SCW03 core' instead of listing each sampling depth.
Lines 113-118: I suggest to rephrase a bit... 'according to the following steps: ~1g of sediment… 24h; the siliceous… matter; the treated… analysis in light microscopy (LM; Eclipse... Japan); 'Photomicrographs' or 'Micrographs' instead of 'photographs'.
Line 122: '(FE-SEM; MIRA 3… Czech Republic)' instead of '(FE-SEM) (MIRA 3… Czech Republic)'.
Line 124: 'Morphological analyses of diatoms were performed…' instead of 'Dimensions of the diatoms were measured…'.
RESULTS
I have a question: are all the diatoms that you observed equally well-preserved or not? If so, do you find some trends in the preservation of diatoms along the core?
Lines 127-135: please rephrase. I suggest: "A total of 87 diatom species belonging to 52 different genera were identified… in Korea (Table 2); of these, 6 species, namely… have never been described before in this area.'
Table 2: I suggest to better explain the results summarized in the table… First of all, I suggest to use the term 'occurrence' instead of 'appearance'. Secondly, I am not sure to have clearly interpreted your data: if the black squares indicate the occurrence of a specific taxon (what you unusually define 'cell appearance'), does this mean that where black squares are not reported such taxon is absent? Or do you report the black squares only for those taxa whose ecology is unknown? If so, also the colored cells of the table, and not only those marked by the black square, indicate the occurrence of a specific taxon (in this case with a known ecology), but you should clearly state this in the legend. Thirdly, what is the meaning of the color changes along the depth profile (e.g., Cyclotella litoralis)? Finally, a suggestion: why not to indicate what are the dominant species of the assemblage? I do suggest to count all of your specimens, but just to indicate the most abundant ones.
Figures 2-11 and 12-20: the figures are excellent, but I strongly suggest to rename them using letters instead of numbers (therefore, you will have Fig. 2, composed of A to L micrographs, and Fig. 3, composed of A to I micrographs). Moreover, use the term 'light microscope' or its abbreviation 'LM', not both (otherwise I do not see the benefit of the abbreviations). I suggest to rephrase the legends, e.g. '(A-B) Cymatonitzschia marina (arrow points to raphe); (C-F) Fallacia hodgeana (in C arrow points to raphe; in F arrowheads point to the terminal openings)...'.
Lines 146-227 and 231-315: please check carefully all the descriptions. I note many typos (some names are correctly italicized, but some are not; moreover, I suggest to use the present tense in the descriptions, not the past).
DISCUSSION
Lines 317-368: this is the most problematic section of the paper, and must be carefully rephrased. You have an interesting dataset that is not age-constrained. If this paper is focused on 'subfossils' you must make at least a small effort in order to provide a temporal perspective to the reader. Maybe I am wrong, but you cannot simply state that something occurred 'in the past' (line 352). Try to better valorize your data, at least attempting a reasonable estimation of the age of the sediments (by means of the literature that you cite, for example). Conversely, you should avoid unreliable (even worldwide!) correlations such as those reported at lines 328 and 334.
Lines 370-410: why you report these long descriptions in the discussion section? I better see them in the results section. Then, if there are some environmental or evolutionary implications you can discuss them.
Author Response
The manuscript submitted by Park and collegues and titled 'Identification of New Sub-fossil Diatoms and Flora on the Sediments of Suncheonman Bay, Korea', provides interesting results about the Korean diatom record, that can be relevant for further studies concerning past climate reconstructions. Therefore, the manuscript surely deserve to be published in the Journal of Marine Science and Engineering. However, I strongly suggest some major revisions, listed below.
I am convinced that the authors will be able to address the points highlighted. Then, I will be glad to read once again this interesting paper.
Response: We appreciate for reviewer with such a nice review. We have changed the manuscript as review’s suggestion and all changes can read by ‘Display of Changes’. And we added ‘chronological data’ which of most important comments in review.
Title
Point 1: Why do you cite the term 'Flora'? In the manuscript you only describe the diatom content of the sediment core, not other floral elements. Maybe 'Diatom flora', but not 'Diatoms and Flora'.
Be careful to use the term 'on' ('on the Sediments'): if I correctly understood, you found the diatoms 'in' the sediments.
Response 1: Thank you for review’s advice. We corrected the title.
Revised title: Identification of New Sub-fossil Datom Flora in the Sediments of Suncheonman Bay, Korea
Abstract
Point 2: Lines 21-23: What is the difference between 'fossil' and 'sub-fossil' diatoms? As far as I remember a sub-fossil is a biological remain that was not involved in all the fossilization steps (e.g., no diagenesis), but I am not completely sure. I suggest to define it more clearly and to use just one of the two terms throughout the manuscript. But the major problem is: how old is your diatom assemblage? Without any age constraint, how can you apply your nice findings for past climate reconstructions? You should try to address this point in the manuscript. Even if you cannot provide absolute ages you can give some estimates…
Response 2: Thank you for review’s important point. We totally agree with your opinion about insert age information. That is the reason, we added chronological data (Figure 2 and Table 2) and defied sub-fossil diatom in discussion before 4.1 section. Also, we inserted about chronological data in materials and methods and result (section 2.2 Analysis of Chronology; 3.1 Sedimentary facies analysis)
Keywords
Point 3: Line 29: 'new record' is probably better than 'new recorded'.
Response 3: Thank the correct words. We accepted your advice.
Introduction
Point 4: Line 38: what is the meaning of 'geologically transient'? Moreover, I do not find a clear reason for the use of the term 'therefore' immediately after the previous sentence, since there is no logical connections between geologically transient environments and their productivity.
Response 4: Thank you for review’s advice. We deleted this sentence.
Point 5: Lines 41-42: do you mean that estuaries provide a 'sink' for the excess of organic matter produced and consumed at the land-to-sea transition zone? I suggest to rephrase the sentence…
Response 5: Thank you for review’s point out. We have rewrote the sentence and hope review can understand it.
Moreover, they filter out pollutants supplied to the ocean and provide a place to excessive nutrients consumption by estuary organisms
Point 6: Lines 59-60: add 'them' after 'making'; add 'favorable' after 'habitat'; delete 'ranging'; and specify what are the macrobenthic and the macrorganisms you cite. Why note to specify the most important species of macrobenthic and macroorganisms involved? It is just a minor suggestion…
Response 6: We appreciate review’s grammar correcting. We have changed the sentence as review’s recommendation.
In particular, the natural environment and ecosystems within Suncheonman Bay are well preserved, making them a habitat favorable for many species ranging from marine organisms
Point 7: Lines 60-62: I suggest to better emphasize the role of diatoms. On my opinion is better to state 'In these regions, primary producers like diatoms play a critical role as a food source for large invertebrates and fishes'.
Response 7: Thank you for straighten out of the sentence. We have changed the sentence as review’s suggestion.
Point 8: Lines 63-67: I suggest to rephrase the first sentence in this way: 'Diatoms are unicellular algae characterized by a biomineralized (opaline) cell wall that may fossilize and be preserved in the sedimentary record.' Then you should emphasize that 'Diatoms thrives in very different environments (e.g., hot springs, polar regions, fresh, brackish and marine waters) and are extremely sensitive to the physical and chemical changes (e.g. temperature, salinity and nutrients) of the waters." Then, on my personal opinion you must highlight the logical link between the two sentences, for example stating 'Therefore, fossil diatoms represent an excellent source of information about past climate change and its effect on the aquatic ecosystems'.
Response 8: We accepted review’s opinion, the sentences has been changed as review’s recommendation.
Diatoms are unicellular algae characterized by a biomineralized (opaline) cell wall that may fossilize and be preserved in the sedimentary record [16,17].Diatoms thrives in very different environments (e.g., hot springs, Polar Regions, fresh, brackish and marine waters) and extremely sensitive to the physical and chemical changes (e.g., temperature, salinity and nutrients) of the waters [17-22]. Therefore, fossil diatoms represent and excellent source of information about past climate change and its effect on the aquatic ecosystems.
Point 9: Lines 69-72: not necessary, you already state at lines 60-62 that diatoms are critical in these environments.
Response 9: We agree with review’s opinion. The sentence deleted
Point 10: Lines 72-73: not necessary.
Response 10: We agree with review’s opinion. The sentence deleted.
Point 11: Lines 72-78: not necessary and wrong in many parts. Note that the Cretaceous ranges from 145 to 65 Ma, that diatom probably originated in the Triassic, and most importantly, that the fossil diatom record is not at all restricted to the Miocene (that lasted from around 23 to 5 Ma). I suggest to delete this passage.
Response 11: Thank you for review’s comment. We decided to remove that parts.
Point 12: Line 83: what is a 'sedimentary' diatom? Fossil (or subfossil if you prefer) is ok, but I never heard the term 'sedimentary diatoms'.
Response 12: Thank you for review’s advice. We deleted ‘sedimentary’ and fixed the sentence.
To date, however, there have been relatively few studies of sub-fossil diatoms on the southern Korean coast.
Point 13: Lines 90-91: please rephrase, not clear… 'has been used to freshwater flow'???
Response 13: We tried to rewrite the sentence, please read the new sentence if reviewer understand or not
Among the studies conducted to date, the investigation of phytoplankton community in Dong Cheon River and Isa Cheon River stream into Suncheonman bay was the most interested
Point 14: Lines 92-94: please rephrase. I suggest: 'In this study we describe a newly recorded sub-fossil diatom assemblage recovered in the sediments of the Suncheonman Bay'.
Response 14: Thank you for review’s advice. We have changed the sentence as review’s organized sentence on the manuscript.
Materials and Methods
Point 15: Line 99: 'lateral' instead of 'horizontal'; 'sediments' instead of 'sedimentary layers'.
Response 15: Thank you for review’s advice. We corrected the words as review’s recommendation.
Drilling to identify the vertical and lateral changes in sediments layers was carried out using a peat core sampler with a 52 mm sample diameter (Peat Sampler, Eijkelkamp Soil & Water, Netherlands)
Point 16: Lines 101-102: how these cores correlates among each others? I suggest to provide some pictures of these cores, highlighting their most relevant sedimentological features (e.g., color, sedimentary structures etc.).
Response 16: Thank you for review’s advice. Actually figure 1. was our mistake. It should appear SCW03 only. Thus, we have corrected figure 1. Moreover, we added Figure 2 which the results of dating and photographs of core SCW03
Point 17: Line 105: see above. Where is this 'sedimentological description'?
Response 17: Thank you for review’s point out. We have added ‘sedimentological description in result 3.1 with figure 2.
Point 18: Lines 106-107: what do you mean for 'analysis of chronology'?; what do you mean for 'sediment layer was well preserved'?
Response 18: Thank you for review’s advice. We added chronological data and Sedimentological description on Table 2, Figure 2, materials and methods, Result 3.1. And for clear meaning, we also rewrite a sentence ‘Shells in the SCW03 core were selected for the analysis of chronology and diatoms because the sediment layer was well preserved.’
Point 19: Table 1: 'Location' instead of 'Address'; I think 'Depth' is better than 'Altitude'.
Response 19: Thank you for review’s advice. We have changed the words as review’s comment. And we removed SCW01 and SCW02 to correct our mistake. Actually SCW01 and SCW02 should drop out in this manuscript.
Point 20: Lines 111-112: 'were collected every 0.5 m along the SCW03 core' instead of listing each sampling depth.
Response 20: Thank you for review’s nice idea. We accepted your opinion, and replaced the words.
Thirteen samples of diatoms were collected every 0.5 m along the SCW03 core.
Point 21: Lines 113-118: I suggest to rephrase a bit... 'according to the following steps: ~1g of sediment… 24h; the siliceous… matter; the treated… analysis in light microscopy (LM; Eclipse... Japan); 'Photomicrographs' or 'Micrographs' instead of 'photographs'.
Response 21: We appreciate review’s advice. We have changed as review’s comment
Their analysis was conducted according to the following steps: 1 g of sediment was dried at 60 °C for 24 h; the siliceous material was boiled with 20 mL of 30% hydrogen peroxide (H2O2) and washed with distilled water to remove organic matter; the treated samples were mounted with Pleurax (Mountmedia, Wako, Japan) and briefly heated using an alcohol lamp for subsequent analysis using a light microscope (LM; Eclipse Ni, Nikon, Japan) equipped with Nomarski differential interference contrast optics (DIC). Photomicrographs were taken using a digital camera (DS-Ri2, Nikon, Japan).
Point 22: Line 122: '(FE-SEM; MIRA 3… Czech Republic)' instead of '(FE-SEM) (MIRA 3… Czech Republic)'.
Response 22: Thank you review’s advice. We have changed the form as review’s comment
The membranes were placed on stubs and coated with gold-palladium for analysis using a field emission scanning electron microscope (FE-SEM; MIRA 3, TESCAN, Czech Republic).
Point 23: Line 124: 'Morphological analyses of diatoms were performed…' instead of 'Dimensions of the diatoms were measured…'.
Response 23: Thank you review’s advice. We have changed the words as review’s comment
Morphological analyses of diatoms were performed using ImageJ v1.32 software [51]. Taxonomical nomenclature was based on recent taxonomic information guidelines [52].
Results
Point 24: I have a question: are all the diatoms that you observed equally well-preserved or not? If so, do you find some trends in the preservation of diatoms along the core?
Response 24: Thank you for review’s very important question. Yes. We observed the diatoms equally well-preserved. And quantitative analyse also have done. However, the diatom quantitative data, more detailed geology, chronology and environmental interpret will submit follow-up paper. As the result, we observed specific pattern of diatom appearance.
Point 25: Lines 127-135: please rephrase. I suggest: "A total of 87 diatom species belonging to 52 different genera were identified… in Korea (Table 2); of these, 6 species, namely… have never been described before in this area.'
Response 25: Thank you review’s advice. We clarify the sentence as review’s recommendation.
A total of 87 diatom species belonging to 52 different genera were identified in the sediments from Suncheonman Bay in Korea (Table 2); of these, 6 species, namely, Cymatonitzschia marina, Fallacia hodgeana, Navicula mannii, Metascolioneis tumida, Surirella recedens, and Thalassionema synedriforme have never been described before in this area.
Point 26: Table 2: I suggest to better explain the results summarized in the table… First of all, I suggest to use the term 'occurrence' instead of 'appearance'. Secondly, I am not sure to have clearly interpreted your data: if the black squares indicate the occurrence of a specific taxon (what you unusually define 'cell appearance'), does this mean that where black squares are not reported such taxon is absent? Or do you report the black squares only for those taxa whose ecology is unknown? If so, also the colored cells of the table, and not only those marked by the black square, indicate the occurrence of a specific taxon (in this case with a known ecology), but you should clearly state this in the legend. Thirdly, what is the meaning of the color changes along the depth profile (e.g., Cyclotella litoralis)? Finally, a suggestion: why not to indicate what are the dominant species of the assemblage? I do suggest to count all of your specimens, but just to indicate the most abundant ones.
Response 26: Thank you for review’s important point out. We tried to re-write about table 3 (original manuscript: table 2) to recognize the information of occurrence diatoms easily.
- We added results summarize in table legend. The sentence of 'A total of 72 diatom species belonging to 52 different genera were identified’ and ‘(6 speices; Cymatonitzschia marina, Fallacia hodgeana, Navicula mannii, Metascolioneis tumida, Surirella recedens, and Thalassionema synedriforme)’ were added
- We have changed the word of ‘appearance’ to ‘occurrence’.
- Black square: We tired remove all of the colour and added black square where the cell appearance depth.
- Colour changes: Originally two colour means marine and freshwater habitats (e.g., Cyclotella litoralis). As review’s comment, the reader can confused. Thus, we decided remove all colour and added habitat cell on the table.
- Dominant species assemblage: It is related response 24. We already analysed quantitative diatom species in SCW03. Thus, we can present abundant species in SCW03 core. However the data of quantitative diatoms, geology, chronology and environmental condition will submit follow-up paper. Also, if we add the dominant species in this manuscript, the direction of this paper could be change. This is the reason, we did not present quantitative diatom data in this manuscript. Even though, do reviewer want to insert dominant species information?
Point 27: Figures 2-11 and 12-20: the figures are excellent, but I strongly suggest to rename them using letters instead of numbers (therefore, you will have Fig. 2, composed of A to L micrographs, and Fig. 3, composed of A to I micrographs). Moreover, use the term 'light microscope' or its abbreviation 'LM', not both (otherwise I do not see the benefit of the abbreviations). I suggest to rephrase the legends, e.g. '(A-B) Cymatonitzschia marina (arrow points to raphe); (C-F) Fallacia hodgeana (in C arrow points to raphe; in F arrowheads point to the terminal openings)...'.
Response 27: Thank you for review’s advice. We have changed figure legend as you advised in figure 3, 4 (original figure number was 2-11, 12-20; new figure 2 has been inserted).
Figure 3. Light microscope (A, G, H) and Field emission scanning electron microscope (Fe-SEM)(B-F, I-J) micrographs of diatoms: (A-B) Cymatonitzschia marina (arrow points to raphe), (C-F) Fallacia hodgeana (in C arrow points to raphe; in F arrowheads point to the terminal poenings), (G-J) Navicula mannii with raphe and ribbon-shaped areola (arrow points to raphe and ribbon shaped areola). Scale bar=10 µm.
Figure 4. Fe-SEM micrographs of diatoms. (A-B) Metascolioneis tumida (arrow points to raphe), (C-D) Surirella recedens (arrow points to raphe), (E-I) Thalassionema synedriforme. Scale bar=10 µm.
Point 28: Lines 146-227 and 231-315: please check carefully all the descriptions. I note many typos (some names are correctly italicized, but some are not; moreover, I suggest to use the present tense in the descriptions, not the past).
Response 28: Thank you for review’s advice. We changed italicized at a species and replaced the present tense in all description. Review can find them by ‘display of changes’
Discussion
Point 29: Lines 317-368: this is the most problematic section of the paper, and must be carefully rephrased. You have an interesting dataset that is not age-constrained. If this paper is focused on 'subfossils' you must make at least a small effort in order to provide a temporal perspective to the reader. Maybe I am wrong, but you cannot simply state that something occurred 'in the past' (line 352). Try to better valorize your data, at least attempting a reasonable estimation of the age of the sediments (by means of the literature that you cite, for example). Conversely, you should avoid unreliable (even worldwide!) correlations such as those reported at lines 328 and 334.
Response 29: Thank you for review’s advice. We have added chronological data Table 2 and Figure 2. for readers. Additionally, we analysed the data of quantitative diatoms, but we need more advanced data and discusses. We will submit follow-up paper.
Point 30: Lines 370-410: why you report these long descriptions in the discussion section? I better see them in the results section. Then, if there are some environmental or evolutionary implications you can discuss them.
Response 30: Thank you for review’s advice. We constructed a section of ‘remark of raphe’ in result. All raphe descriptions moved to remark of raphe sections in each species. Thus roles, environmental and evolutionary contents of raphe remains in the discussion part.
Round 2
Reviewer 1 Report
Since this manuscript is specific for a special issue on taxonomy, the scope qualifies my second choice mentioned in the previous review report. Under this scenario, i.e. from the point of view of a pure research article on taxonomy, I agree with the other reviewer's comment to publish it in JSME, and currently I don't have any further comments.
And the revisions done in this version are appreciated.
Author Response
Since this manuscript is specific for a special issue on taxonomy, the scope qualifies my second choice mentioned in the previous review report. Under this scenario, i.e. from the point of view of a pure research article on taxonomy, I agree with the other reviewer's comment to publish it in JSME, and currently I don't have any further comments.
And the revisions done in this version are appreciated.
Response: We appreciate review's comments and read the manuscript again. We tired to corrected the manuscript as review 2 comments.
Reviewer 2 Report
Dear Editor, dear Authors,
the revised version of the manuscript now titled 'Identification of New Sub-fossil Diatoms in the Sediments of Suncheonman Bay, Korea' has been significantly improved. It is now almost ready for the publication, but after moderate to minor revisions, mostly related to the structure of the sentences and to small typos. I reported below the major points to address.
Sincerely
ABSTRACT
Lines 23-24 - I suggest to rewrite: 'One sedimentary core has been extracted in 2018'
INTRODUCTION
Lines 38-39 - I suggest to rephrase as follow: 'Estuaries and wetlands are among the most productive aquatic ecosystems, providing a home for both freshwater and marine plants, and a source of nutrients for a variety of animal communities adapted to brackish waters'
Line 40 - Delete 'that have' before 'adapted'
Lines 41-42 - I suggest to delete from 'and provide' to 'organisms'
Line 60-61 - I suggest to rephrase as follow: 'a habitat favorable for many species of marine organisms'
Line 69 - add 'are' after 'and'
Line 71 - 'an', not 'and'!
Lines 86-90 - If I correctly understood the meaning, I suggest to rephrase as follow: 'There have been relatively few studies on sub-fossil diatoms along the Southern Korean coast. Marine to brackish sub-fossil diatom assemblages were initially studied in the Pohang and Gampo sediments in 1975, then they were extended to the regions of Bukpyeong and Pohang in the East Sea and in the regions of Mankyung-Dongin river estuary, Dodaecheon River, Ilsan estuary, Chollipo, Isanpo, Sabsi-do and Kunsan in the Yellow Sea.' IMPORTANT POINT: the authors take for grant that the reader already know the difference between fossil and subfossil… But the reader will clearly understand such difference only in the Discussions (lines 414-418). Why not to define the difference in the Introduction? For example at line 68, in this way: '(…) in the sedimentary record. The sub-fossil diatoms herein described consist in Holocene diatom remains not fully involved in the fossilization process'. If you accept such modification, delete the definition that you provide in the Discussions. The alternative is to avoid to define a ‘sub-fossil’. Indeed, in all the papers on subfossil diatoms that I read after I received this manuscript, the term ‘sub-fossil’ is never defined…
Line 96 – ‘Fishes’ instead of ‘fish’
Line 99 – ‘was the most complete’ or ‘was the most interesting’ instead of ‘was the most interested’
Line 109-110 – Indeed, if you do not cite and describe the other cores, the lateral changes of sedimentary facies are no more relevant herein. I suggest to rephrase as follow: ‘Drilling was carried out using a peat core sampler (52 mm diameter; Peat Sampler…)’
Line 116 – ‘Shells’… Of what?!
Figure 2 – Panel A: why do you report 2 scales (‘Elevation’ and ‘Depth’)? It’s confusing… Depth is enough! Panel C: it is truncated, while there is a complete one floating together with table 2 at page 8… Probably a typo, but check it.
Table 2 – ‘Shells’… Of what?!
Lines 152-168 – Sorry, I appreciated your efforts aimed at provide a sedimentological description, but this is full of typos and unclear in many passages. Try to rephrase, describe and interpret each facies, from A to C (you do not describe facies B!), with order. I give you some suggestions below:
3.1 Sedimentary facies analysis
153 The core SCW03 is comprised mostly consists of greenish-grey silty clay, and can be divided distinguished into three
154 sedimentary facies based on the according to color, fossils content, and sedimentary structures (Figure 2A and
155 C). Facies A is characterized by the yellowish brown mottling structures. Shell or shell
156 fragments (OF WHAT???) are not observed in this facies (so, for sake of simplicity, do not cite them herein). In Facies B, the yellowish brown mottling
157 structures are decreased less abundant and shell (OF WHAT???) fragments are observed occur sporadically. The size of sShell (OF WHAT???)
158 fragments is are about 2 mm in diameter. Facies C is represented by highly concentrated shells (OF WHAT???)
159 and shells (OF WHAT???) fragments in several, centimeters thick intervals(?).
160 The sediments of core SCW03 is have been interpreted as deposited in a tidal flat (REFERENCES?). It means that the observation of the shells (OF WHAT???)
161 or shell fragments (OF WHAT???) and the silty clay deposits are not significantly different from the
162 current environment (THIS SENTENCE IS CONFUSING!). The increase in the Abundant mottling structures in facies A indicates a change
163 to an environment where oxidation is relatively dominant an oxygen-rich environment, indicating that the sea level
164 has gradually decreased slightly (THIS PASSAGE IS UNCLEAR, PROBABLY YOU MEAN THAT A RELATIVE DECREASE OF SEA LEVEL PROMOTED THE OXYGENATION, BUT PLEASE REPHRASE...). In the meantime, no change in sedimentary facies at the
165 top and bottom of facies C indicates that there has been a temporary change in the
166 environment (i.e. storm, flood, etc.) (THIS PASSAGE IS UNCLEAR, PLEASE REPHRASE).
167 In the meantime, the results of age dating for five samples in the core SCW03 shows
168 a range from 1170 to 1560 cal. yr BP (Table 2 and Figure 2B) (THIS IS NOT A SEDIMENTARY FACIES INTERPRETATION).
Line 185 – I suggest: ‘Occurrence (black squares) and habitat of diatom species by depths’
Line 186 – ‘species’ not ‘speices’
Line 188 – Delete ‘Black square mark shows cell appearances’
Table 3 – Please carefully check the diatoms species names… For example, at line 49 you reported ‘Haslae’ instead of ‘Haslea’; at line 72 ‘Surirella receden’ instead of ‘Surirella recedens’.
Line 201 – ‘photomicrographs’ instead of ‘photographs’
Line 202 – ‘openings’ instead of ‘poenings’
Descriptions of Cymatonitzschia marina – I suggest some corrections, encouraging the authors to make a further effort and apply a similar approach to the other descriptions (that overall are very good, but must be carefully checked):
Cymatonitzschia marina (F.W.Lewis) Simonsen 1974 (Figures 23,. A-B 3)
207 Basionym: Cymatopleura marina F.W.Lewis 1861 [114105]
208 Synonym: Cymatopleura marina F.W.Lewis 1861 [114105]
209 Original description: Simonsen 1974: 56, pl. 41: figs 5-9 [115106]
210 Description: Cells (valves?) are observed to be solitary, usually lying in the valve view.
211 Valves are linear lanceolate, with very acute ends. Valves are strictly isopolar,
212 and not constricted in the middle. Overall dimensions included a Valve length ranges from 58.42 to
213 67.94 μm and valve width from 9.14 to 11.28 μm. The valve faces have numerous
214 undulations (9–11), with a distance between two undulations in the range of 4.81–7.97 μm.
215 Undulations are found to have a nearly trapezoidal shape (Figure 3A). The valve
216 surface has irregular punctate on the undulations. A raphe system was are is observed
217 to be running around one side of the valve margin. Striae uUniseriate striae are found to be
218 densely spaced, with approximately 28–29 per 10 μm observed, on one side of the valve
219 margin (Figure 3B3, arrow)
220 Depth occurringence in the core: 2.0 m.
221 Distribution: This species hasve been reported from brackish water or to marine waters environment
222 mainly [5968-7364]. This taxon has been reported and from some estuariesy, e.g. East River, New
223 York and Long Island Sound [116107]. Cymatopleura marina was first recorded from the
224 Indian Ocean [115106].
225 Differential diagnosis: This genus differs from Cymatopleura. The genus
226 Cymatopleura, as a member of the Surirellaceae, has a completely different raphe
227 morphology, which runs along the edge of the valve around the entire margin, whereas
228 in Cymatonitzschia it is, like in Nitzschia, limited to one of the sides side [115106]
229 Remarks onf the raphe system: Cymatonitzschia marina has an eccentric keeled raphe placed near
230 one margin side [108156]. This means that the raphe shows (?) through the edge of the valve
231 around the entire marine body (??? rephrase), and it appears on one of the sides [115106].
Lines 419-460 – Sorry, the discussion is still not convincing… Please, make a bit effort. I suggest some corrections below, BUT PLEASE ALWAYS CONSIDER A CRUCIAL VARIABLE, I.E. THE SEDIMENT ACCUMULATION RATE. THIS MEANS THAT WITHOUT CLEAR CHRONOLOGICAL CONSTRAINTS BASED ON 14C OR OTHER PROXIES, THE COMPARISONS AMONG SITES VERY FAR FROM EACH OTHER AND JUST BASED ON THE DEPTH (THAT IS DEPENDENT ON THE SEDIMENT ACCUMULATION RATES!) ARE MERELY SPECULATIVE! IN OTHER WORDS, CAUTION IS NEEDED IN COMPARING YOUR RECORD WITH THE UKRAINIAN AND DUTCH ONES JUST ON THE BASIS OF THE DIATOM DEPTH OCCURRENCE… CONSIDER TO AVOID SUCH COMPARISONS FOR NOW, DISCUSSING ONLY YOUR RECORD.
In this study, the analysis of sub-fossil diatoms among core samples obtained from
421 Suncheon Bay, Korea, was carried out. Within the SCW03 core sample, a total of 52 genera
422 and 87 species of subfossil diatoms were identified; detected, with locations ranging from the surface to
423 within 6 m of the basement and among them, six species of newly recorded sub-fossil diatom never recorded before have been
424 found. At a depth of 4.0 m, the maximum variety of diatom samples was observed, with
425 23 genera 34 species; while depths of 0.5 m and 3.5 m revealed the lowest variety of
426 diatoms, with 12 genera 14 species and 10 genera 14 species, respectively (Table 32). Highest and lowest taxonomic richness occur at a depth of 4.0 m (23 genera, 34 species) and 3.5 m (10 genera, 14 species), respectively (Table 3).
427 Among the diatoms observed, Giffenia sp. appeared occurs in at all sections depths ranging from depths of
428 0.1 m to 6.0 m. WHAT DOES THIS IMPLY? UNFORTUNATELY YOU DO NOT REPORT ANY REFERENCE ABOUT ITS HABITAT, SINCE YOU DID NOT REACH THE SPECIES LEVEL, BUT PROBABLY YOU CAN SPECULATE A BIT ABOUT THE ECOLOGICAL MEANING OF THIS GENUS… IF YOU CAN’T,YOU SHOULD STATE THAT FURTHER INVESTIGATIONS ARE NEEDED IN ORDER TO BETTER UNDERSTAND THE ‘PALEOENVIRONMENTAL’ MEANING OF THIS DIATOM. In addition, to Amphora sp., Auliscus sculptus and Fragilaria capucina were
429 found only at 0.1 m depths, and therefore are hypothesized to they have only recently entered the Suncheon Bay. DO YOU THINK THAT THIS MAY BE RELATED, FOR EXAMPLE, TO ANTHROPOGENIC ACTIVITIES? SUGGEST SOME INTERPRETATIONS IF YOU CAN!
430 In particularInterestingly, A. sculptus was found in the 15–54 cm sections only in the surface sediments extracted from of the Karkinit Bay
431 sediment in Ukraine, possibly confirming a recent introduction of this diatom in the estuarine environments (HOW THE AUTHORS THAT WORKED ON THE KARKINIT BAY INTERPRETED THE OCCURRENCE OF A. sculptus?), and it is which represents a relatively recently discovered fossil diatom,
432 which was not observed in deeper samples [28142]. However, additional chronological
433 studies of each sample have to be conducted to determine whether A. sculptus appeared
434 in these two locations, far from each other namely Suncheonman Bay (South Korea) and Karkinit Bay (Ukraine),
435 around the same time [28142]. On the other hand, F. capucina was found deeper (7-11 m) in the sediment
436 between 7 to 11 meters in of the De Waai lake sedimentary soil in the (Netherlands), and A.
437 sculptus, has been mostly mainly found in the shallow cores at shallow depth between depths of (15-and 54 cm) [28142,30143],. The
438 opposite was observed in contrast with the Suncheonman Bay sedimentary soil [30143]. Cymatosira
439 lorenziana, Fallacia hodgeana, Pleurosigma sp., Semiorbis sp., and Trachyneis aspera no longer
440 observed since it appeared only occur at 6.0 m depth. It is hypothesized that this is due to climate,
441 environmental, or topographical changes in the Suncheonman Bay habitat. Intriguingly, C. lorenziana is
442 mainly found in warm waters, whereas T. aspera is mainly found in the is mainly distributed in the Antarctic area,
443 with almost opposite habitat characteristics; therefore, their co-occurrence is therefore puzzling and deserve future investigations the causative environmental
444 changes in Suncheon Bay could not be identified in this study [144148-150146]. However,
445 identifying past environmental changes in Suncheon Bay is an important source of
446 information for the prediction of future environmental changes, and should be
447 investigated further.
448 Most of the marine and brackish species appeared within occur at a depth of 5.0 to 6.0
449 m of the SCW03 core sample, and the emergence while of freshwater species gradually increased
450 within at a depth ranges of 3.0 to 4.5 m. Interestingly, only marine and brackish species
451 appeared at in the 2.5 m section (Figure Table 32). Marine and some freshwater species
452 appeared within occur the range of at 1.0 m to 2.0 m depths, while only marine and brackish species appeared
453 at 0.5 m depth,; and finally, freshwater species increased again at 0.1 m depths. These Cchanges in the flora of diatoms assemglage composition
454 observed suggests that the sediments recovered from SCW03 represents were deposited in a marine area which had little scarcely influenced by fresh water
455 inflow in the past (3.0-4.5 m depth; AGE?) but and experienced an renewed influx of fresh water in recent times (0.1 m depth)., which increased around the
456 3.0 to 4.5 m depth range and decreased in 0.5 to 2.5 m range. Due to this influx, the number
457 of freshwater species has increased in the 0.1 m section, suggesting that there was an
458 increase in recent freshwater inflow (Table 2-3, Figure 2) [151,152147,148]. For more
459 accurate results Solid interpretations of the triggers responsible for such environmental changes, however, require further studies require comprehensive environmental change evaluation,
460 including a quantitative analysis of diatoms and a more accurate chronological analysis of core samples.
Lines 463-464 - 'In this study we identified six species up to now unrecord in the Suncheonman Bay area' is better than 'We found six unrecorded species'; Italicize 'Thalassionema synedriforme'
Lines 470-471 - Please rephrase, not clear at all.
Lines 473-514 - Sorry, but it is still unclear to me why do you put so much emphasis on the raphe system. This final paragraph does not add relevant information (you already provided a good description before). Consider to delete.
Author Response
Response to Reviewer 2 Comments
The revised version of the manuscript now titled 'Identification of New Sub-fossil Diatoms in the Sediments of Suncheonman Bay, Korea' has been significantly improved. It is now almost ready for the publication, but after moderate to minor revisions, mostly related to the structure of the sentences and to small typos. I reported below the major points to address.
Sincerely
Response: Thank you very much of review’s comments. We tried to rewrite some sentences and fixed words as review’s advices. And, tried to discuss only our described data. In this reason, some paragraphs have been added, the other paragraph has been deleted. All changes marked red characters.
Abstract
Point 1: Lines 23-24 - I suggest to rewrite: 'One sedimentary core has been extracted in 2018'
Response 1: Thank you for review’s advice. We have replaced the words as review’s recommendation.
Introduction
Point 2: Lines 38-39 - I suggest to rephrase as follow: 'Estuaries and wetlands are among the most productive aquatic ecosystems, providing a home for both freshwater and marine plants, and a source of nutrients for a variety of animal communities adapted to brackish waters'
Response 2: Thank you for review’s advice. We have rephrase as review’s advice.
Point 3: Line 40 - Delete 'that have' before 'adapted'
Response 3: Thank you for review’s advice. We have rephrase as review’s advice.
Point 4: Lines 41-42 - I suggest to delete from 'and provide' to 'organisms'
Response 4: Thank you for review’s advice. We have delete as review’s advice
Original sentence: Moreover, they filter out pollutants supplied to the ocean and provide a place to excessive nutrients consumption by estuary organisms [4,6,7 ].
New sentence: Moreover, they filter out pollutants supplied to the ocean [4,6,7].
Point 5: Line 60-61 - I suggest to rephrase as follow: 'a habitat favorable for many species of marine organisms'
Response 5: ok
Original sentence: In particular, the natural environment and ecosystems within Suncheonman Bay are well preserved, making them a habitat favorable for many species ranging from marine organisms [15].
New Sentence: In particular, the natural environment and ecosystems within Suncheonman Bay are well preserved, making them a habitat favorable for many species of marine organisms [15]
Point 6: Line 69 - add 'are' after 'and'
Response 6: Thank you for review’s advice. We have corrected as review’s advice
New sentence: Diatoms thrives in very different environments (e.g., hot springs, Polar Regions, fresh, brackish and marine waters) and are extremely sensitive to the physical and chemical changes (e.g., temperature, salinity and nutrients) of the waters
Point 7: Line 71 - 'an', not 'and'!
Response 7: Thank you for review’s advice. We replaced ‘an’ as review’s advice
New sentence: Therefore, fossil diatoms represent an excellent source of information about past climate change and its effect on the aquatic ecosystems.
Point 8: Lines 86-90 - If I correctly understood the meaning, I suggest to rephrase as follow: 'There have been relatively few studies on sub-fossil diatoms along the Southern Korean coast. Marine to brackish sub-fossil diatom assemblages were initially studied in the Pohang and Gampo sediments in 1975, then they were extended to the regions of Bukpyeong and Pohang in the East Sea and in the regions of Mankyung-Dongin river estuary, Dodaecheon River, Ilsan estuary, Chollipo, Isanpo, Sabsi-do and Kunsan in the Yellow Sea.
Response 8-1: Thank you for review’s advice. We rephrased as review’s advice
' IMPORTANT POINT: the authors take for grant that the reader already know the difference between fossil and subfossil… But the reader will clearly understand such difference only in the Discussions (lines 414-418). Why not to define the difference in the Introduction? For example at line 68, in this way: '(…) in the sedimentary record. The sub-fossil diatoms herein described consist in Holocene diatom remains not fully involved in the fossilization process'. If you accept such modification, delete the definition that you provide in the Discussions. The alternative is to avoid to define a ‘sub-fossil’. Indeed, in all the papers on subfossil diatoms that I read after I received this manuscript, the term ‘sub-fossil’ is never defined…
Response 8: Thank you for review’s point out. We totally agree with review’s opinions. So, we added a sentence in introduction (The sub-fossil diatoms herein described consist in Holocene diatom remains not fully involved in the fossilization process.) and deleted a paragraph in discussion about sub-fossil part.
Point 9: Line 96 – ‘Fishes’ instead of ‘fish’
Response 9: Thank you for review’s advice. We corrected the word
Point 10: Line 99 – ‘was the most complete’ or ‘was the most interesting’ instead of ‘was the most interested’
Response 10: Thank you for review’s advice. We replaced ‘was the most interesting’ as review’s advice
Original sentence: Among the studies conducted to date, the investigation of phytoplankton community in Dong Cheon River and Isa Cheon River stream into Suncheonman bay was the most interested
New sentence: Among the studies conducted to date, the investigation of phytoplankton community in Dong Cheon River and Isa Cheon River stream into Suncheonman bay was the most interesting
Point 11: Line 109-110 – Indeed, if you do not cite and describe the other cores, the lateral changes of sedimentary facies are no more relevant herein. I suggest to rephrase as follow: ‘Drilling was carried out using a peat core sampler (52 mm diameter; Peat Sampler…)’
Response 11: Thank you for review’s advice. We rephrased as review’s advice
New sentence: Drilling was carried out using a peat core sampler (52 mm diameter; Peat Sampler, Eijkelkamp Soil & Water, Netherlands).
Point 12: Line 116 – ‘Shells’… Of what?!
Response 12: Thank you for review’s point out. We changed ‘shells’ to ‘shell fragments’ in manuscript. Actually, we found particles of shells in the core sample. Thus, we could not identified species of shells. If review wanted to know about species of shell, unfortunately we do not know now and need more research.
Point 13: Figure 2 – Panel A: why do you report 2 scales (‘Elevation’ and ‘Depth’)? It’s confusing… Depth is enough! Panel C: it is truncated, while there is a complete one floating together with table 2 at page 8… Probably a typo, but check it.
Response 13: Thank you for review’s advice. We have changed figure 2 and we described more in 3.1 and 3.2.
Point 14: Table 2 – ‘Shells’… Of what?!
Response 14: Thank you for review’s point out. We changed ‘shells’ to ‘shell fragments’ in table 2. Also, please see the response 12.
Point 15: Lines 152-168 – Sorry, I appreciated your efforts aimed at provide a sedimentological description, but this is full of typos and unclear in many passages. Try to rephrase, describe and interpret each facies, from A to C (you do not describe facies B!), with order. I give you some suggestions below:
Response 15: Thank you for review’s comments. We renewal the paragraphs ‘3.1 Sedimentary facies analysis’ and added ‘3.2 age dating’. We tried to describe each facies from A to C in 3.1.
3.1 Sedimentary facies analysis
Point 16: 153 The core SCW03 is comprised mostly consists of greenish-grey silty clay, and can be divided distinguished into three sedimentary facies based on the according to color, fossils content, and sedimentary structures (Figure 2A and C). Facies A is characterized by the yellowish brown mottling structures. Shell or shell fragments (OF WHAT???) are not observed in this facies (so, for sake of simplicity, do not cite them herein). In Facies B, the yellowish brown mottling structures are decreased less abundant and shell (OF WHAT???) fragments are observed occur sporadically. The size of sShell (OF WHAT???) fragments is are about 2 mm in diameter. Facies C is represented by highly concentrated shells (OF WHAT???) and shells (OF WHAT???) fragments in several, centimeters thick intervals(?).
The sediments of core SCW03 is have been interpreted as deposited in a tidal flat (REFERENCES?). It means that the observation of the shells (OF WHAT???)
161 or shell fragments (OF WHAT???) and the silty clay deposits are not significantly different from the
162 current environment (THIS SENTENCE IS CONFUSING!). The increase in the Abundant mottling structures in facies A indicates a change
163 to an environment where oxidation is relatively dominant an oxygen-rich environment, indicating that the sea level
164 has gradually decreased slightly (THIS PASSAGE IS UNCLEAR, PROBABLY YOU MEAN THAT A RELATIVE DECREASE OF SEA LEVEL PROMOTED THE OXYGENATION, BUT PLEASE REPHRASE...). In the meantime, no change in sedimentary facies at the
165 top and bottom of facies C indicates that there has been a temporary change in the
166 environment (i.e. storm, flood, etc.) (THIS PASSAGE IS UNCLEAR, PLEASE REPHRASE).
167 In the meantime, the results of age dating for five samples in the core SCW03 shows
168 a range from 1170 to 1560 cal. yr BP (Table 2 and Figure 2B) (THIS IS NOT A SEDIMENTARY FACIES INTERPRETATION).
Response 16: Thank you for review’s sincere comments. We tried to rephrase these paragraphs and tried to separate ‘sedimentary facies analysis’ and ‘age dating’. Also tried to change sentence as review’s recommendation.
Point 17: Line 185 – I suggest: ‘Occurrence (black squares) and habitat of diatom species by depths’
Response 17: Thank you for review’s advice. We have changed as review’s recommendation.
New sentence: Table 3. Occurrence (black squares) and habitat of diatom species by depth.
Point 18: Line 186 – ‘species’ not ‘speices’
Response 18: Thank you for correction of wrong spells. We have corrected the wrong spells to ‘species’
Point 19: Line 188 – Delete ‘Black square mark shows cell appearances’
Response 19: Thank you for review’s kind comment. We corrected all legend of table 3 as review’s advice from point 17 to 19.
New legend: Table 3. Occurrence (black squares) and habitat of diatom species by depth. A total of 72 diatom species belonging to 52 different genera were identified. Star marks on the specific name are newly recorded species in Korea (6 species; Cymatonitzschia marina, Fallacia hodgeana, Navicula mannii, Metascolioneis tumida, Surirella recedens, and Thalassionema synedriforme).
Point 20: Table 3 – Please carefully check the diatoms species names… For example, at line 49 you reported ‘Haslae’ instead of ‘Haslea’; at line 72 ‘Surirella receden’ instead of ‘Surirella recedens’.
Response 20: Thank you for review’s indicating the important problems. We checked all spells of table 3 diatom species.
Point 21: Line 201 – ‘photomicrographs’ instead of ‘photographs’
Response 21: Thank you for review’s advice. We replaced ‘photomicrographs’ in figure 3 and figure 4 legends.
New sentence: Figure 4. Field emission scanning electron microscopic photomicrographs of diatoms. (A-B) Metascolioneis tumida (arrow points to raphe), (C-D) Surirella recedens (arrow points to raphe), (E-I) Thalassionema synedriforme. Scale bar=10 µm.
Point 22: Line 202 – ‘openings’ instead of ‘poenings’
Response 22: Thank you for review’s point out the wrong spell. We corrected it.
New sentence: Figure 3. Light microscope (A, G, H) and Field emission scanning electron microscopic (B-F, I-J) photomicrographs of diatoms: (A-B) Cymatonitzschia marina (arrow points to raphe), (C-F) Fallacia hodgeana (in C arrow points to raphe; in F arrowheads point to the terminal openings), (G-J) Navicula mannii with raphe and ribbon-shaped areola (arrow points to raphe and ribbon shaped areola). Scale bar=10 µm.
Point 23: Descriptions of Cymatonitzschia marina – I suggest some corrections, encouraging the authors to make a further effort and apply a similar approach to the other descriptions (that overall are very good, but must be carefully checked):
Cymatonitzschia marina (F.W.Lewis) Simonsen 1974 (Figures 23,. A-B 3)
207 Basionym: Cymatopleura marina F.W.Lewis 1861 [114105]
208 Synonym: Cymatopleura marina F.W.Lewis 1861 [114105]
209 Original description: Simonsen 1974: 56, pl. 41: figs 5-9 [115106]
210 Description: Cells (valves?) are observed to be solitary, usually lying in the valve view. Valves are linear lanceolate, with very acute ends. Valves are strictly isopolar, and not constricted in the middle. Overall dimensions included a Valve length ranges from 58.42 to 67.94 μm and valve width from 9.14 to 11.28 μm. The valve faces have numerous undulations (9–11), with a distance between two undulations in the range of 4.81–7.97 μm.
215 Undulations are found to have a nearly trapezoidal shape (Figure 3A). The valve surface has irregular punctate on the undulations. A raphe system was are is observed to be running around one side of the valve margin. Striae uUniseriate striae are found to be densely spaced, with approximately 28–29 per 10 μm observed, on one side of the valve margin (Figure 3B3, arrow)
Depth occurringence in the core: 2.0 m.
221 Distribution: This species hasve been reported from brackish water or to marine waters environment mainly [5968-7364]. This taxon has been reported and from some estuariesy, e.g. East River, New York and Long Island Sound [116107]. Cymatopleura marina was first recorded from the Indian Ocean [115106].
225 Differential diagnosis: This genus differs from Cymatopleura. The genus Cymatopleura, as a member of the Surirellaceae, has a completely different raphe morphology, which runs along the edge of the valve around the entire margin, whereas in Cymatonitzschia it is, like in Nitzschia, limited to one of the sides side [115106]
229 Remarks onf the raphe system: Cymatonitzschia marina has an eccentric keeled raphe placed near
Response 23: Thank you for review’s comments. We tried to fix all review’s recommendation and other species also matched formats.
230 one margin side [108156]. This means that the raphe shows (?) through the edge of the valve
Response: Thank you for review’s advice. That is exact meaning of review’s understand. But, we have rewrote the sentence for readers understand.
Original sentences: Cymatonitzschia marina has an eccentric keeled raphe placed near one margin side [108]. This means that the raphe shows through the edge of the valve around the entire marine body, and it appears on one of the sides [106].
New sentences: Cymatonitzschia marina has an eccentric keeled raphe placed through the edge of the valve, and it appears on one of the sides [106,108].
231 around the entire marine body (??? rephrase), and it appears on one of the sides [115106].
Response: (previous response is related) Thank you for review’s point out. We have rewrote that sentences.
Point 24: Lines 419-460 – Sorry, the discussion is still not convincing… Please, make a bit effort. I suggest some corrections below, BUT PLEASE ALWAYS CONSIDER A CRUCIAL VARIABLE, I.E. THE SEDIMENT ACCUMULATION RATE. THIS MEANS THAT WITHOUT CLEAR CHRONOLOGICAL CONSTRAINTS BASED ON 14C OR OTHER PROXIES, THE COMPARISONS AMONG SITES VERY FAR FROM EACH OTHER AND JUST BASED ON THE DEPTH (THAT IS DEPENDENT ON THE SEDIMENT ACCUMULATION RATES!) ARE MERELY SPECULATIVE! IN OTHER WORDS, CAUTION IS NEEDED IN COMPARING YOUR RECORD WITH THE UKRAINIAN AND DUTCH ONES JUST ON THE BASIS OF THE DIATOM DEPTH OCCURRENCE… CONSIDER TO AVOID SUCH COMPARISONS FOR NOW, DISCUSSING ONLY YOUR RECORD.
In this study, the analysis of sub-fossil diatoms among core samples obtained from Suncheon Bay, Korea, was carried out. Within the SCW03 core sample, a total of 52 genera and 87 species of subfossil diatoms were identified; detected, with locations ranging from the surface to within 6 m of the basement and among them, six species of newly recorded sub-fossil diatom never recorded before have been found. At a depth of 4.0 m, the maximum variety of diatom samples was observed, with 23 genera 34 species; while depths of 0.5 m and 3.5 m revealed the lowest variety of diatoms, with 12 genera 14 species and 10 genera 14 species, respectively (Table 32). Highest and lowest taxonomic richness occur at a depth of 4.0 m (23 genera, 34 species) and 3.5 m (10 genera, 14 species), respectively (Table 3).
Among the diatoms observed, Giffenia sp. appeared occurs in at all sections depths ranging from depths of 0.1 m to 6.0 m. WHAT DOES THIS IMPLY? UNFORTUNATELY YOU DO NOT REPORT ANY REFERENCE ABOUT ITS HABITAT, SINCE YOU DID NOT REACH THE SPECIES LEVEL, BUT PROBABLY YOU CAN SPECULATE A BIT ABOUT THE ECOLOGICAL MEANING OF THIS GENUS… IF YOU CAN’T,YOU SHOULD STATE THAT FURTHER INVESTIGATIONS ARE NEEDED IN ORDER TO BETTER UNDERSTAND THE ‘PALEOENVIRONMENTAL’ MEANING OF THIS DIATOM.
Response: Thank you for review’s advice. We did not impart a meaning to Giffenia sp. To be honest, Giffenia sp. possibly new species. Now Giffenia sp. known to one species in the word, morphologic features are different with Giffenia cocconeiformis. But we did not described in the manuscript. So, the sentence of ‘Giffenia sp. appeared occurs in at all sections depths ranging from depths of 0.1 m to 6.0 m’ have been deleted.
In addition, to Amphora sp., Auliscus sculptus and Fragilaria capucina were found only at 0.1 m depths, and therefore are hypothesized to they have only recently entered the Suncheon Bay. DO YOU THINK THAT THIS MAY BE RELATED, FOR EXAMPLE, TO ANTHROPOGENIC ACTIVITIES? SUGGEST SOME INTERPRETATIONS IF YOU CAN!
Response: Thank you for Review’s advice. We could not find any interaction of three diatom species and anthropogenic activities. And if we want to know about the diatoms and anthropogenic activities, chronological data and historical information should needed. We are not sure even if some references could cover it.
430 In particularInterestingly, A. sculptus was found in the 15–54 cm sections only in the surface sediments extracted from of the Karkinit Bay sediment in Ukraine, possibly confirming a recent introduction of this diatom in the estuarine environments (HOW THE AUTHORS THAT WORKED ON THE KARKINIT BAY INTERPRETED THE OCCURRENCE OF A. sculptus?), and it is which represents a relatively recently discovered fossil diatom,
432 which was not observed in deeper samples [28142]. However, additional chronological studies of each sample have to be conducted to determine whether A. sculptus appeared in these two locations, far from each other namely Suncheonman Bay (South Korea) and Karkinit Bay (Ukraine), around the same time [28142]. On the other hand, F. capucina was found deeper (7-11 m) in the sediment between 7 to 11 meters in of the De Waai lake sedimentary soil in the (Netherlands), and A. sculptus, has been mostly mainly found in the shallow cores at shallow depth between depths of (15-and 54 cm) [28142,30143],.
The opposite was observed in contrast with the Suncheonman Bay sedimentary soil [30143]. Cymatosira lorenziana, Fallacia hodgeana, Pleurosigma sp., Semiorbis sp., and Trachyneis aspera no longer observed since it appeared only occur at 6.0 m depth. It is hypothesized that this is due to climate, environmental, or topographical changes in the Suncheonman Bay habitat. Intriguingly, C. lorenziana is mainly found in warm waters, whereas T. aspera is mainly found in the is mainly distributed in the Antarctic area, with almost opposite habitat characteristics; therefore, their co-occurrence is therefore puzzling and deserve future investigations the causative environmental changes in Suncheon Bay could not be identified in this study [144148-150146]. However, identifying past environmental changes in Suncheon Bay is an important source of information for the prediction of future environmental changes, and should be investigated further.
448 Most of the marine and brackish species appeared within occur at a depth of 5.0 to 6.0 m of the SCW03 core sample, and the emergence while of freshwater species gradually increased within at a depth ranges of 3.0 to 4.5 m. Interestingly, only marine and brackish species appeared at in the 2.5 m section (Figure Table 32). Marine and some freshwater species appeared within occur the range of at 1.0 m to 2.0 m depths, while only marine and brackish species appeared at 0.5 m depth,; and finally, freshwater species increased again at 0.1 m depths. These Cchanges in the flora of diatoms assemglage composition observed suggests that the sediments recovered from SCW03 represents were deposited in a marine area which had little scarcely influenced by fresh water inflow in the past (3.0-4.5 m depth; AGE?) but and experienced an renewed influx of fresh water in recent times (0.1 m depth)., which increased around the 3.0 to 4.5 m depth range and decreased in 0.5 to 2.5 m range. Due to this influx, the number of freshwater species has increased in the 0.1 m section, suggesting that there was an
increase in recent freshwater inflow (Table 2-3, Figure 2) [151,152147,148].
For more accurate results Solid interpretations of the triggers responsible for such environmental changes, however, require further studies require comprehensive environmental change evaluation, including a quantitative analysis of diatoms and a more accurate chronological analysis of core samples.
Response 24: Thank you for review’s very important point. We discussed about review’s advice and decided to delete the comparison of Ukraine and Netherlands paragraph. And, we rewrite words and sentences as review’s recommendation marked red color.
Deleted part: In particular interestingly, A. sculptus was found in the 15–54 cm sections only in the surface sediments extracted from of the Karkinit Bay sediment in Ukraine. Possibly confirming a recent introduction of this diatom in the estuarine environments, and it is which represents a relatively recently discovered fossil diatom, which was not observed in deeper samples [144]. However, additional chronological studies of each sample have to be conducted to determine whether A. sculptus appeared in these two locations, far from each other namely Suncheonman Bay (South Korea) and Karkinit Bay (Ukraine), around the same time [144]. On the other hand, F. capucina was found deeper (7-11 m) in the sediment of De Waai lake sedimentary soil (Netherlands), and A. sculptus, has been mostly mainly found in shallow depth (15 and 54 cm) [144,145]. The opposite was observed in contrast with the Suncheonman Bay sedimentary soil [147].
Point 25: Lines 463-464 - 'In this study we identified six species up to now unrecord in the Suncheonman Bay area' is better than 'We found six unrecorded species'; Italicize 'Thalassionema synedriforme'
Response 25: Thank you for review’s advice. We fixed as review’s comments.
New sentence: In this study we identified six species up to new record in Suncheonman Bay area Cymatonitzschia marina, Fallacia Hodgeana, Navicula mannii, Metascolioneis tumida, Surirella recedens, and Thalassionema synedriforme.
Point 26: Lines 470-471 - Please rephrase, not clear at all.
Response 26: Thank you for review’s comment. We tried to rewrite the sentence, hope review and reader can understand easily
Original sentence: This study is significant because it is possible to identify if the extinction of unrecorded species when diatom diversity studies are conducted in modern life
New sentence: This study is such a meaningful. Because if we study modern composition of diatoms in Suncheonman Bay or Korea, we could estimate unrecorded 6 diatoms have been extinction in this area.
Point 27: Lines 473-514 - Sorry, but it is still unclear to me why do you put so much emphasis on the raphe system. This final paragraph does not add relevant information (you already provided a good description before). Consider to delete.
Response 27: Thank you review’s opinions. We are agree with review’s idea. Raphe system makes confusing of subjects of the manuscript. Thus, we decided delete the paragraph ‘4.3 Raphe system’.
